# MAViL: Masked Audio-Video Learners

**Po-Yao Huang**[1] **Vasu Sharma**[1] **Hu Xu**[1] **Chaitanya Ryali**[1] **Haoqi Fan**[1] **Yanghao Li**[1]
**Shang-Wen Li**[1] **Gargi Ghosh**[1] **Jitendra Malik**[1,2] **Christoph Feichtenhofer**[1]

[1]FAIR, Meta          [2]University of California, Berkeley

## Abstract

We present Masked Audio-Video Learners (MAViL) to learn audio-visual representations with three complementary forms of self-supervision: (1) reconstructing masked raw audio and video inputs, (2) intra-modal and inter-modal contrastive learning with masking, and (3) self-training to predict aligned and contextualized audio-video representations learned from the first two objectives. Empirically, MAViL achieves state-of-the-art audio-video classification performance on AudioSet (53.3 mAP) and VGGSound (67.1% accuracy), surpassing recent self-supervised models and supervised models that utilize external labeled data. Notably, pre-training with MAViL not only enhances performance in multimodal classification and retrieval tasks, but it also improves the representations of each modality in isolation, without relying on information from the other modality during uni-modal fine-tuning or inference. The code and models are available at https://github.com/facebookresearch/MAViL.

## 1 Introduction

We study self-supervised learning (SSL) from audio and video, two rich, heterogeneous, yet closely related modalities of human perception. There are two primary forms of self-supervision commonly used: reconstruction and contrastive learning. By reconstructing masked text tokens on large-scale corpora, BERT [20] pre-training has achieved groundbreaking results in various NLP tasks. Masked autoencoders (MAEs) have recently emerged as powerful tools for learning uni-modal representations in various modalities such as image [37], video [27], and audio [41]. They employ an asymmetric encoder-decoder architecture with a substantial portion of encoder inputs being masked, resulting in efficient uni-modal SSL. Additionally, contrastive learning has been widely used to learn coordinated multimodal representations for image-text [75, 93] and audio-visual [67, 71] tasks.

In this work, we develop a novel approach that integrates MAE and contrastive learning to enhance audio-video representation learning for both audio-video and audio-only tasks. This integration poses unique challenges under the multimodal paradigm, necessitating meticulous model design to effectively manage diverse multimodal inputs in MAE, while also adapting contrastive learning to accommodate masking of the majority of inputs for efficiency. Our method, Masked Audio-Video Learners (MAViL), comprises a pair of encoders to encode audio and video, a fusion encoder, and distinct decoders tailored for reconstructing raw inputs or contextualized representations in each modality (as illustrated in Fig. 1). We design MAViL with three types of objectives outlined next.

Firstly, we extend uni-modal MAE to multimodal and utilize a fusion encoder to exchange information from all modalities. MAViL reconstructs raw inputs that has been removed under a high masking ratio (*e.g.* 80%). With this, it learns *complementary* audio-video representations by reconstructing a single modality input, with the supplementary context from the other. Secondly, we adapt contrastive learning [34, 85], with the majority of uni-modal inputs being masked, to learn an *aligned* audio-video latent space efficiently. MAViL employs two types contrastive objectives: (i) An *inter-modal* contrast that brings together paired video and audio clips from the same video and contrasts them with other

37th Conference on Neural Information Processing Systems (NeurIPS 2023).

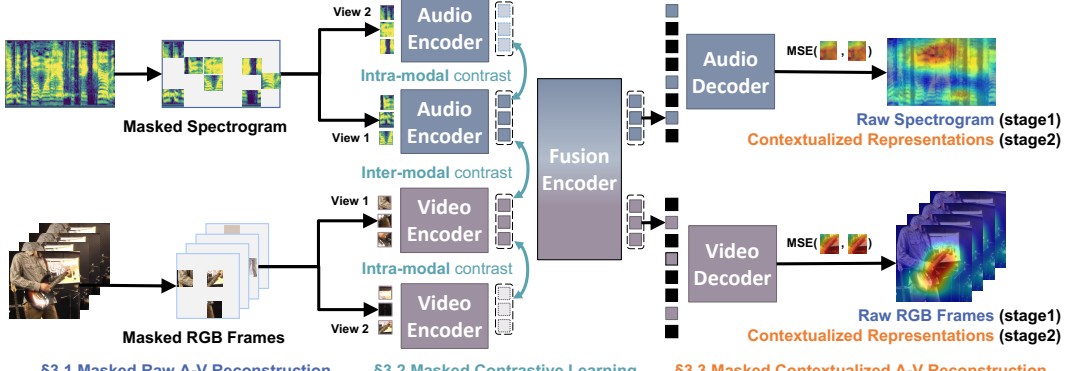

Figure 1: **Masked Audio-Video Learners (MAViL)** exploit three objectives for learning representations from audio video pairs with masking: (1) Raw audio-video reconstruction. (2) Inter-modal and intra-modal contrastive learning with masking. (3) Reconstructing aligned and contextualized audio-video representations via student-teacher learning. (Please see Fig. 2 for details.)

unpaired clips sampled in a mini-batch. (ii) An *intra-modal* contrast that draws closer the two masked views of the same audio or video, while pushing away other samples from the same modality.

Thirdly, unlike conventional MAEs that focus on reconstructing *heterogeneous* and *raw* inputs (audio *or* video), we propose a novel pre-training task that reconstructs *homogeneous* and *contextualized* audio-video representations in a joint (audio-*and*-video) latent space (illustrated in Fig. 2).

We are motivated by the recent successes in visual SSL that use disparate teacher models to generate contextualized representations as reconstruction targets. For example, BEiT [8] uses features by DALL-E [77] as its target. MaskFeat [91] predicts masked target features such as HOG [19] or DINO [10]. Inspired by these uni-modal teachers, we take a step further to propose a new masked *contextualized audio-video reconstruction* pretext task. Our core idea is: instead of predicting raw inputs in heterogeneous modalities, the model with masked-view inputs jointly predicts contextualized audio-video representations in the homogeneous (aligned) latent space generated by a teacher (*i.e.* an identical model pre-trained with the masked reconstruction and contrastive objectives above) with complete-view inputs. This approach ensures the students learn well-aligned and contextualized audio-video representations and thus improves their performance on downstream tasks. To achieve this, without relying on external teacher models, we employ a simple two-stage self-training framework (illustrated in Fig. 2). In stage-1, we train the teacher MAViL and use raw inputs as its reconstruction targets. In stage-2, the student MAViL (the final model for later fine-tuning) learns to reconstruct the aligned and contextualized audio-video representations generated by the MAViL teacher.

Our experimental results confirm MAViL's superiority in audio-video classification (AudioSet-20K, AudioSet-2M, VGGSounnd) and retrieval tasks (VTT and YouCook) where it surpasses the best self-supervised and supervised pre-trained models by large margins. Notably, MAViL not only learns strong joint audio-video representations, but can also improve single modality encoders, without using the other modality it pre-trained with during fine-tuning or inference.

In summary, MAViL makes the following contributions for learning self-supervised audio-video representations: (1) Efficient contrastive learning for both intra-modal and inter-modal context via masking. (2) Introducing a new pretext task for multimodal MAE that predicts aligned and contextualized audio-video representations, surpassing the performance of reconstructing uni-modal raw inputs as used in conventional uni-modal/multimodal MAE. (3) Achieving new state-of-the-art performance in 7 audio-visual classification, audio-visual retrieval tasks, and (4) audio-only tasks under the SSL setup without using labeled data for pre-training.

## 2   Related Work

**Supervised Audio-Video Models.** The connection between visual signals and co-occurring audio context for language acquisition and world comprehension in infants [80] has motivated the research of audio-visual learning [76, 18]. Previous studies have explored audio-visual ASR [14, 73] and

person identification [1] prior to the deep learning era. More recently, there has been significant attention on audio-video representation learning for classification [65, 46, 22, 43, 95, 64]. However, these supervised approaches rely on abundant labeled data, which are costly and time-consuming to obtain. While efforts have been made to create large-scale labeled datasets [35], the dependency on extensive annotation hinders progress in audio-video modeling. In contrast, MAViL focuses on self-supervised learning of robust audio-video representations without the need for labeled data.

**Self-Supervised Audio-Video Representation Learning.** To exploit abundant unlabeled video and audio content on the Internet and reduce annotation efforts, self-supervised techniques have been explored for learning audio-video representations [4, 2, 3, 48, 82]. Inter-modal contrastive learning is a widely used approach that learns to associate paired audio-video clips as self-supervision [57, 67, 63, 71]. Techniques such as data augmentation [70, 87] and harder negatives mining [98, 78, 62] have been studied to improve its performance. MAViL unifies masked autoencoding and contrastive learning. We notice concurrent and independent study CAV-MAE [32] and AV-MAE [29] recently explored joint masked autoencoding to reconstruct raw audio and video. Unlike CAV-MAE, MAViL takes a step further to incorporate intra-modal contrastive learning from two masked views. Most importantly, MAViL employs student-teacher learning to reconstruct contextualized and aligned representations instead of raw inputs as in CAV-MAE and AV-MAE. These innovations lead to MAViL's superior performance.

**Student-teacher learning** or knowledge distillation (KD) [40, 84, 69] was originally developed for model compression [60, 17], aiming to transfer knowledge from a larger teacher model to a smaller student model. In the context of SSL, KD has recently gained attention for improving the distribution of representations by distilling [93] or by reconstructing contextualized targets from a teacher model [33, 8, 92]. Approaches like MoCo [36], DINO [10], and dBOT [53] utilize self-training to bootstrap targets from previous model snapshots during pre-training. Data2vec [7] performs self-training in disparate modalities, where each one bootstraps complete-view contextualized targets independently. In contrast, to our best knowledge, MAViL is the first work that employs masked multimodal context for self-training. It uniquely incorporates complete-view multimodal fusion in the teacher model, while the student model receives masked-view inputs.

## 3 Masked Audio-Video Learners

We introduce Masked Audio-Video Learners (MAViL), a self-supervised audio-video representation learning framework (see Fig. 1). MAViL consists of two stages: In stage-1, MAViL jointly reconstructs raw spectrograms and RGB frames by exploiting complementary information from each modality (§3.1), and couples this with contrastive learning to encourage alignment between semantically similar instances, both *within* and *across* modalities. (§3.2). In stage-2 (see Fig. 2), we employ either the stage-1 model trained with raw targets (in iter. 1) or the last trained stage-2 model (in iter. 2+) as the teacher for self-training. We use the teacher's aligned and contextualized audio-video representations, obtained with complete-view inputs, to guide the student with masked-view inputs (§3.3). In the following, we provide details of these three types of self-supervision, starting with MAViL-stage1 trained with raw inputs.

### 3.1 Masked Raw Audio-Video Reconstruction

Human perception involves processing visual and acoustic context jointly. In line with this, MAViL utilizes multimodal Transformers to fuse and exploit the complementary information from both audio and video. It aims to reconstruct audio and video simultaneously as self-supervision, which sets it apart from uni-modal MAE approaches such as MAE [37], Audio-MAE [41], or Video-MAE [27].

Given a raw audio-video pair $(a, v) \in \mathcal{D}$, we begin by patchifying and tokenizing raw audio spectrograms and video frames into audio and video tokens. This involves applying (audio/video) transforms, followed by 2D/3D-convolutions and flattening. This process embeds raw inputs into $\mathbf{a} = [a_1 \ldots a_N]$ audio spectrogram tokens and $\mathbf{v} = [v_1 \ldots v_M]$ video tokens, where $a_i, v_j \in \mathbb{R}^H$. To incorporate positional information, similar to MAE, we employ fixed 2D sinusoidal positional embeddings with the embedded tokens for each modality. We then randomly mask the majority (*i.e.* 80%) of audio and video tokens. Only the remaining 20% unmasked audio ($\mathbf{a}'$) and video ($\mathbf{v}'$) tokens

are respectively fed into the audio ($f_\mathrm{a}(.)$) and video ($f_\mathrm{v}(.)$) Transformer encoders. This process results in uni-modal embeddings, denoted as $\mathbf{a}_\mathrm{um} = f_\mathrm{a}(\mathbf{a}')$ and $\mathbf{v}_\mathrm{um} = f_\mathrm{v}(\mathbf{v}')$.

Following the uni-modal encoders, we incorporate a multimodal *fusion* encoder denoted as $g_\mathrm{av}(.)$ to model multimodal context. We explore two variants for this purpose: Vanilla Transformers [86] and Multimodal Bottleneck Transformers (MBT) [64]. For vanilla Transformers, we jointly encode the audio and video tokens by: $(\mathbf{a}_\mathrm{um}^{l+1} \| \mathbf{v}_\mathrm{um}^{l+1}) = \mathrm{Transformer}^l(\mathbf{a}_\mathrm{um}^l \| \mathbf{v}_\mathrm{um}^l)$, where "$\|$" represents concatenation. Details of MBT implementation is in Appendix. In both variants, we stack $L$ Transformer layers to obtain the jointly encoded top-layer outputs $\mathbf{a}_\mathrm{mm}$ and $\mathbf{v}_\mathrm{mm}$.

For reconstruction, we employ vanilla Transformer blocks as the audio $f_\mathrm{a}^{-1}(.)$ and video $f_\mathrm{v}^{-1}(.)$ decoders. The fusion encoder's outputs ($\mathbf{a}_\mathrm{mm}$ and $\mathbf{v}_\mathrm{mm}$) are firstly projected and padded with trainable [MASK] tokens. After restoring the original order (time-frequency for audio and space-time for video tokens), we add the decoders' (fixed 2-D sinusoidal) positional embeddings and input the restored sequences into the decoders. At the top of the decoders, we incorporate linear heads to reconstruct the raw inputs. Specifically, the decoder outputs for spectrogram reconstruction are denoted as $\hat{\mathbf{a}} = f_\mathrm{a}^{-1}(g_\mathrm{av}(f_\mathrm{a}(\mathbf{a}')))$ and for RGB frame reconstruction as $\hat{\mathbf{v}} = f_\mathrm{v}^{-1}(g_\mathrm{av}(f_\mathrm{v}(\mathbf{v}')))$. For notation clarity, we omit the [MASK] tokens and linear projection head. Let $\hat{\mathbf{a}}_i, \mathbf{a}_i^\mathrm{raw} \in \mathbb{R}^{H_\mathrm{raw}^\mathrm{a}}; i = 1 \ldots n$ denote the audio decoder's output and the ground truth reference of the $i$-th masked spectrogram patch. Similarly, $\hat{\mathbf{v}}_j, \mathbf{v}_j^\mathrm{raw} \in \mathbb{R}^{H_\mathrm{raw}^\mathrm{v}}; j = 1 \ldots m$ for video patches. In masked audio-video reconstruction, MAViL is self-supervised by minimizing the mean squared error (MSE) loss $\mathcal{L}_r^\mathrm{raw}$ defined as:

$$\mathcal{L}_r^\mathrm{raw} = \frac{1}{n} \sum_{i=1}^{n} (\hat{\mathbf{a}}_i - \mathbf{a}_i^\mathrm{raw})^2 + \frac{1}{m} \sum_{j=1}^{m} (\hat{\mathbf{v}}_j - \mathbf{v}_j^\mathrm{raw})^2. \tag{1}$$

## 3.2 Contrastive Audio-Video Learning with Masking

Contrastive learning is widely used to learn uni-modal [13, 36, 15] and multimodal [75, 71] representations by aligning multiple "views" of the same instance. These views can be either *within*-modality observations of the instance itself (e.g., the same audio under different volumes) or semantic observations *across* modalities (e.g., a video and its corresponding audio). MAViL utilizes InfoNCE [85] loss for contrastive learning. Let $\mathbf{x} = [\mathbf{x}_1 \ldots \mathbf{x}_B], \mathbf{y} = [\mathbf{y}_1 \ldots \mathbf{y}_B]; \mathbf{x}_i, \mathbf{y}_j \in \mathbb{R}^H$ be the instance-level representations of audio/video in a batch of size $B$. The contrastive loss $\mathcal{L}_\mathrm{c}(\mathbf{x}, \mathbf{y})$ is defined as:

$$\mathcal{L}_\mathrm{c}(\mathbf{x}, \mathbf{y}) = -\frac{1}{B} \sum_{i=1}^{B} \log \frac{\exp(\mathrm{S}(\mathbf{x}_i, \mathbf{y}_i)/\tau)}{\sum_{j=1}^{B} \exp(\mathrm{S}(\mathbf{x}_i, \mathbf{y}_j)/\tau))}, \tag{2}$$

where $\mathrm{S}(\mathbf{x}_i, \mathbf{y}_j) = \frac{\mathbf{x}_i^T \mathbf{y}_j}{\|\mathbf{x}_i\| \|\mathbf{y}_j\|}$ is the cosine similarity between $\mathbf{x}_i, \mathbf{y}_j$ and $\tau$ is the softmax temperature. MAViL employs two types of contrastive losses for self-supervision listed below:

**Inter-modal contrastive learning** facilitates alignments *across* modalities. We first average the sequence of uni-modal encoder outputs[1] as the instance-level representations, namely, $\mathbf{a}_\mathrm{emb} = \mathrm{Avg}(\mathbf{a}_\mathrm{um})$ and $\mathbf{v}_\mathrm{emb} = \mathrm{Avg}(\mathbf{v}_\mathrm{um})$. The positive pairs (the numerator in Eq.(2)) consist of video and audio clips from the same video, while all the other combinations of sampled audio-video pairs are considered negatives (the denominator). Inter-modal contrast encourages the representations of paired audio and video to be closer to each other, while simultaneously pushing away mismatched ones.

**Intra-modal contrastive learning** promotes alignment *within* each modality. It aims to bring the representations of different views, such as different augmented versions of an audio (or video), closer to each other. To achieve this, for each modality, we apply random masking and sample a second view that contains 20% tokens for encoding. The idea is to treat masking as a form of augmentation to generate two contrasting views in the same modality, which has been proven effective in visual SSL [36, 13]. The two views of the same instance are considered as a positive pair, which are then contrasted against the other (negative) combinations of the instances in the same modality. Formally, let $\bar{\mathbf{a}}_\mathrm{emb}$ and $\bar{\mathbf{v}}_\mathrm{emb}$ be the embeddings of the second-view audio and video. MAViL employs the

---

[1]We do not use the fusion output $\mathbf{a}_\mathrm{mm}$ and $\mathbf{v}_\mathrm{mm}$ for contrastive learning since the fusion layer provides a shortcut/leakage of information exchange for every paired audio-video clip. This results in sub-optimal performance for contrastive learning.

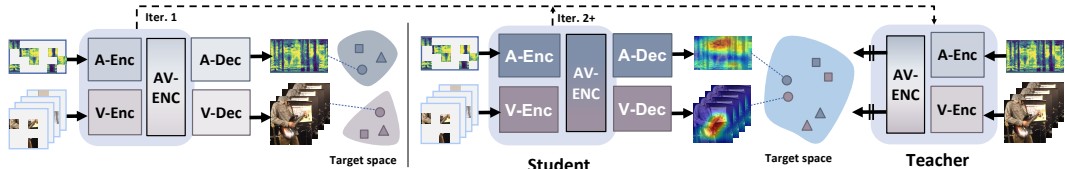

Figure 2: **Masked contextualized audio-video reconstruction** in the joint latent space. Stage-1 (left): Training MAViL with raw inputs as targets. Stage-2 (right): Self-training MAViL student by reconstructing MAViL teacher's aligned and contextualized audio-video representations generated with complete inputs. Repeat stage-2 for $K$ iterations. In the first iteration of stage-2, the stage-1 model is used as the teacher. In subsequent iterations (iteration 2+), the last trained stage-2 student model serves as the new teacher.

inter-modal ($\mathcal{L}_c^{\text{inter}}$) and intra-modal ($\mathcal{L}_c^{\text{intra}}$) contrastive objectives defined as:

$$\mathcal{L}_c^{\text{inter}} = \frac{1}{2}\left[\mathcal{L}_c(\mathbf{a}_{\text{emb}}, \mathbf{v}_{\text{emb}}) + \mathcal{L}_c(\mathbf{v}_{\text{emb}}, \mathbf{a}_{\text{emb}})\right], \quad \mathcal{L}_c^{\text{intra}} = \frac{1}{2}\left[\mathcal{L}_c(\mathbf{a}_{\text{emb}}, \bar{\mathbf{a}}_{\text{emb}}) + \mathcal{L}_c(\mathbf{v}_{\text{emb}}, \bar{\mathbf{v}}_{\text{emb}})\right],$$

(3)

These contrastive losses effectively learn a latent audio-video space where semantically similar instances are close to each other. Note that unlike prior work (*e.g.* [36, 13]), MAViL performs contrastive learning under masked-view. This leads to improved computation efficiency as only a small portion of input tokens are encoded for contrast. And different from CAV-MAE [32], MAViL additionally incorporates intra-modal contrast, which yields superior performance. Overall, let $\alpha$ and $\beta$ be the weights that balance loss terms, MAViL-stage1 is self-supervised by minimizing:

$$\mathcal{L}_{\text{MAViL}} = \mathcal{L}_r^{\text{raw}} + \alpha\mathcal{L}_c^{\text{inter}} + \beta\mathcal{L}_c^{\text{intra}},$$

(4)

### 3.3 Masked Contextualized Audio-Video Reconstruction

To learn robust audio-video representations, we go beyond raw input reconstruction in uni-model MAEs [37, 41, 25], multimodal MAE [6], and CAV-MAE [32]. We propose a new pretext task that reconstructs contextualized audio-video representations in the joint latent space. To achieve this, we employ a simple two-stage training framework illustrated in Fig. 2. In stage-1, we train MAViL with Eq.(4) to reconstruct raw inputs. In stage-2, we employ a student-teacher learning framework for iterative self-training. In the first iteration, the pre-trained MAViL from stage-1 is frozen and serves as the *teacher* model. It generates audio-video representations with complete-view inputs, which are used as the reconstruction targets to guide the re-initialized *student* MAViL. In the subsequent iterations, the last trained stage-2 student MAViL serves as the new teacher.

Formally, we provide the teacher model's encoders with complete-view audio and video inputs to generate aligned and contextualized targets: $\mathbf{a}^{\text{Teacher}}\|\mathbf{v}^{\text{Teacher}} = g_{\text{av}}^{\text{Teacher}}(f_{\text{a}}^{\text{Teacher}}(\mathbf{a})\|f_{\text{v}}^{\text{Teacher}}(\mathbf{v}))$. The student MAViL then learns to reconstruct these contextualized targets with the masked-view inputs. Precisely, $\tilde{\mathbf{a}} = f_{\text{a}}^{-1}(g_{\text{av}}(f_{\text{a}}(\mathbf{a}')))$ for audio and $\tilde{\mathbf{v}} = f_{\text{v}}^{-1}(g_{\text{av}}(f_{\text{v}}(\mathbf{v}')))$ for video, where $\mathbf{a}^{\text{Teacher}}, \mathbf{v}^{\text{Teacher}}, \tilde{\mathbf{a}}, \tilde{\mathbf{v}} \in \mathbb{R}^H$. The contextualized reconstruction objective is defined as:

$$\mathcal{L}_r^{\text{context}} = \frac{1}{n}\sum_{i=1}^{n}(\tilde{\mathbf{a}}_i - \mathbf{a}_i^{\text{Teacher}})^2 + \frac{1}{m}\sum_{j=1}^{m}(\tilde{\mathbf{v}}_j - \mathbf{v}_j^{\text{Teacher}})^2.$$

(5)

In the stage-2 training, we jointly minimize the masked contextualized reconstruction loss and the contrastive loss. The stage-2 (student) MAViL's objective is:

$$\mathcal{L}_{\text{MAViL}} = \mathcal{L}_r^{\text{context}} + \alpha\mathcal{L}_c^{\text{inter}} + \beta\mathcal{L}_c^{\text{intra}}.$$

(6)

Note that Eq.(6) contains pure latent targets[2]. After pre-training, we then fine-tune the audio/video encoders in the final stage-2 MAViL student model in the downstream tasks.

---

[2]Empirically, we found no gain when multi-tasking with stage-1 autoencoding loss on raw pixel/spectrogram.

# 4 Experiments

We performed comprehensive evaluations, including audio-video classification tasks on AudioSet [28] (AS-2M and AS-20K), and VGGSound [11]. Also, we conducted audio-to-video retrieval experiments on MSR-VTT [96] and YouCook [100]. We use AS-20K for model analysis and ablation studies.

## 4.1 Datasets

**AudioSet** contains 2 million 10-second YouTube clips for audio event detection. 527 event types are weakly labeled [51, 38, 39] for each clip, and multiple events can occur in one clip. AudioSet's full training set has two subsets: A class-wise *balanced* (22K clips) and an *unbalanced* (2M clips) set. The *eval* set has 20K clips. We downloaded 1.97M unbalanced training, 20K balanced training, and 19K evaluation clips. We use the full (unbalanced+balanced) training set for pre-training. In the AS-2M task, we fine-tune on the full training set. In the AS-20K task, we fine-tune only on the 20K balanced training set. We report the classification mAP on the 19K *eval* set used by AST [30].

**VGGSound** comprises approximately 200K 10-second video clips annotated with 309 event types that include human actions, sound-emitting objects, *etc*. Unlike AudioSet, VGGSound ensures that an audio event is also visually present in its corresponding clips. VGGSound is divided into 183K training and 15K testing samples. We report top-1 testing classification accuracy.

## 4.2 Implementation Details

MAViL adopts different temporal footprints for audio and video. For audio, following [64, 30], it transforms a 10-second audio under 16K sampling rate into 128 Mel-frequency bands with a 25ms Hanning window shifting every 10ms. The resulting spectrogram is of dimension $1024 \times 128$. MAViL then tokenizes it into non-overlapping $16 \times 16$ patches where both time and frequency have a kernel and stride of 16. The flattened audio tokens have a sequence length $N$ of 512. For video, MAViL processes 4-second clips under 2 frames per second. Each frame has a size of $224 \times 224$. Tokenization is achieved by a 3D convolution, where the spatial kernel and stride are 16, and the temporal kernel and stride are 2. The flattened video tokens have a sequence length $M$ of 784.

Following the design choices in MAE [37], MAViL employs 12-layer Transformers (ViT-B) with 12 attention heads as the encoders for each modality . The embedding dimension $H$ is set to 768. The audio-video fusion encoder layer consists of a 2-layer Transformer (vanilla or MBT) on top of the uni-modal encoders. Similarly, the audio and video decoders utilize 8-layer Transformers with an embedding dimension of 512 and 16 attention heads. MAViL's audio/video encoder and decoder have 86M and 27M parameters, respectively. The floating point operations (FLOPs) for the audio encoder are 48.6G, comparable to the audio encoders in Audio-MAE [41] and CAV-MAE [32].

For pre-training MAViL's audio branch, we randomly initialize it from scratch. For the visual branch, we either randomly initialize it or initialize it with the self-supervised MAE [37] pre-trained on ImageNet (compared in Table 6). Notably, MAViL operates under the fully *self-supervised* setup.

## 4.3 Experimental Setup

We pre-train MAViL on AS-2M without using any of AS-2M labels. We use 80% masking for audio and video. For balancing the losses in Eq.(6), we set $\alpha = 0.1, \tau_c^{inter} = 0.1$ and $\beta = 0.01, \tau_c^{intra} = 1.0$. These hyperparameters scale the gradients from the three losses into a comparable range to improve training stability. We pre-train with 64 V100 GPUs with a 512 accumulated batch size and a 0.0002 learning rate. We pre-train for 20 epochs in stage-1 and in each iteration of stage-2 (for ($K = 3$) iterations). Each session takes 20 hours, resulting in a total pre-training time around 80 hours.

We then evaluate audio and video representations by fine-tuning the pre-trained MAViL in audio-video classification tasks on AS-20K, AS-20M, and VGGSound. We test 3 scenarios: (1) audio-only, (2) video-only, and (3) audio+video. Following MAE [37], we only keep the pre-trained encoders and use the average-pooled top-layer outputs for classification. We adopt the standard fine-tuning pipeline and augmentation in prior audio/audio-video classification works [30, 41, 64]. Specifically, we employ SpecAug [68], mixup [99], balanced sampling [51], and fine-tuning masking [41]. For video, we use standard video augmentations used in video classification [90, 27]. To address the discrepancy in convergence rate between audio and video, we apply a 50% learning rate reduction for the video

encoder during the audio+video (fusion) fine-tuning. Fine-tuning for 100 epochs on AS-2M takes 10 hours. Please refer to Appendix for additional implementation and experimental details.

In the following, we first use AS-20K to analyze MAViL's performance in §4.4. We then present the main comparison to prior works in §4.5. The gray entries represent the shared default setup.

## 4.4 Model Analysis

**Masked Raw Audio-Video Reconstruction.** Table 1 shows the contribution of MAViL's audio-video fusion encoder layer, which exploits complementary multimodal information for reconstructing raw inputs in stage-1. The 'None" (col. 1) indicates the uni-modal MAE baselines without the fusion layer, namely Audio-MAE and Video-MAE under our implementation. The fusion layer, whether it's a vanilla Transformer or MBT, contributes up to 0.4 mAP gains (col. 2-5). While the MBT layers show better results, the improvements is not significant compared to using vanilla Transformers. Also, increasing the depth of the fusion encoder only leads to marginal gains. For simplicity, we default to using a 2-layer vanilla Transformer as the fusion encoder.

| Fusion | None | Vanilla$^{(2)}$ | Vanilla$^{(4)}$ | MBT$^{(4)}$ |
|---|---|---|---|---|
| Audio | 36.4 | $36.8_{(+0.4)}$ | $36.7_{(+0.3)}$ | $36.8_{(+0.4)}$ |
| Video | 17.4 | $17.7_{(+0.3)}$ | $17.6_{(+0.2)}$ | $17.7_{(+0.3)}$ |

Table 1: **Fusion Encoder**: Transformer type$^{(\text{\# layers})}$. Note that the 'None' are the uni-modal MAE baselines (Audio-MAE and Video-MAE). mAP↑ in AS-20K.

| Ratio | 40% | 60% | 70% | 80% | 90% |
|---|---|---|---|---|---|
| Audio | 35.6 | 36.5 | 36.7 | 36.8 | 36.8 |
| Video | 16.8 | 17.3 | 17.5 | 17.7 | 17.5 |

Table 2: **Masking Ratio**. mAP↑ in AS-20K for individual fine-tuning as audio or video classification.

Subsequently, we investigate the masking ratio, a key hyperparameter in masked autoencoding. We use the same masking ratio for audio and video, and ablate different values in Table 2. The results suggest a masking ratio of 80% achieves the best performance. This aligns with the optimal values of 80% in Audio-MAE [41] and 90% in Video-MAE [27]. MAViL encodes only the non-masked tokens significantly reduces the sequence length, resulting in more efficient computation, as the complexity of self-attention layers in Transformers scales quadratically with the sequence length.

**Contrastive Audio-Video Learning with Masking.** Next, we examine the benefits when contrastive learning is used in conjunction with MAE. MAViL performs efficient contrastive learning with only 20% of visible patches in each modality (80% are masked). The results in Table 3 demonstrate the significance of contrastive losses for learning audio-video representations, even with only 20% visible tokens. Unlike CAV-MAE [32] which uses only inter-modal contrast, we observe that both inter-modal and intra-modal contrast are essential, and combining them delivers the best performance.

| Objs. | None | Inter | Intra | Inter+intra |
|---|---|---|---|---|
| Audio | 36.8 | $38.4_{(+1.6)}$ | $38.1_{(+1.3)}$ | $39.0_{(+2.2)}$ |
| Video | 17.7 | $21.0_{(+3.3)}$ | $19.8_{(+2.1)}$ | $22.2_{(+4.5)}$ |

Table 3: **Masked Contrastive Audio-Video Learning**

**Masked Contextualized Audio-Video Reconstruction.** As illustrated in Fig. 2, MAViL employs a two-stage training. Following stage-1 training (Table 3), MAViL's encoders acquire complementary audio-video representations through masked raw audio-video reconstruction, as well as aligned representations through masked contrastive learning. In stage-2, we employ an iterative approach to initialize the student MAViL and train it under masked-views to predict the aligned and contextualized representations generated by the frozen MAViL teacher with full view. The teacher can be either from the stage-1 model (iter. 1) or the last trained stage-2 model (iter. 2 and beyond).

We first compare the reconstruction targets in Table 4a (stage-2, iter. 1). By default MAViL student predicts the last trained MAViL teacher's multimodal fusion encoder outputs (*i.e.*, $g_{\text{av}}^{\text{Teacher}}(f_{\text{a}}^{\text{Teacher}}(\mathbf{a})\|f_{\text{v}}^{\text{Teacher}}(\mathbf{v}))$, M-Fusion, col. 4). The results show that this improves over predicting (col. 1) raw spectrogram/RGB frames (+1.7 audio mAP) and (col. 2) separately pre-trained uni-modal Audio/Video-MAE's outputs (+1.2 audio mAP). The fused multimodal targets (M-Fusion, col. 4) are better (+0.5 audio mAP) than the uni-modal targets (M-Uni, col. 3) (*i.e.*, $f_{\text{a}}^{\text{Teacher}}(\mathbf{a})$ and $f_{\text{v}}^{\text{Teacher}}(\mathbf{v})$ before the fusion encoder).

| Target | Raw | A/V-MAE | M-Uni | M-Fusion |
|--------|-----|---------|-------|----------|
| Audio | 39.0 | $39.5_{(+0.5)}$ | $40.2_{(+1.2)}$ | $40.7_{(+1.7)}$ |
| Video | 22.2 | $23.1_{(+0.9)}$ | $23.8_{(+1.6)}$ | $24.1_{(+1.9)}$ |

| Iter. | None | 1 | 2 | 3 | 4 |
|-------|------|---|---|---|---|
| Audio | 39.0 | $40.7_{(+1.7)}$ | $41.5_{(+2.5)}$ | $41.8_{(+2.8)}$ | $41.8_{(+2.8)}$ |
| Video | 22.2 | $24.1_{(+1.9)}$ | $24.6_{(+2.4)}$ | $24.8_{(+2.6)}$ | $24.9_{(+2.7)}$ |

(a) **Reconstruction Targets** (Stage-2, iteration-1)    (b) **Iterations** (Stage-2)

Table 4: **Targets** and **iterations** in MAViL's stage-2 contextualized audio-video reconstruction. (a) Target comparison in the 1st iteration. Raw: spectrogram/RGB frames. Contextualized: A-MAE/V-MAE (uni-modal MAE output); M-Uni (MAViL's uni-modal output); M-Fusion (MAViL's multimodal fusion output). (b) Iteration comparison. 'None' is the MAViL-stage1 baseline in Table 3.

For the 2nd and beyond iterations of stage-2 training, we leverage the last trained MAViL student as the new teacher. As shown in Table 4b, substantial improvements (asymptotically saturated after the 3rd iteration) are achieved over iterations through masked prediction of aligned and contextualized audio-video representations by the MAViL teacher. The accumulated gains with the proposed contextualized audio-video reconstruction in stage-2 are +2.8 audio and +2.6 video mAP *without* using additional data or external teacher models.

**Additional Ablations.** In Table 5 we ablate several more aspects of MAViL: (a) Pre-training with more data is useful (100% > 50% > 10% of AS-2M) (Tab. 5a). (b) ImageNet (IN) self-supervised pre-training is beneficial for the video encoder (Table 5b). ImageNet supervised pre-training (IN-SL) is useful yet we avoid this to keep MAViL fully self-supervised. (c) Longer pre-training is beneficial and 20 epochs are sufficient (Table 5c). (d) Increasing the encoder sizes (ViT-S/B/L) improves mAP (Table 5d). We use ViT-B for efficiency and fair comparison as default.

| # | 200K | 1M | 2M |
|---|------|-----|-----|
| A | 34.1 | 40.5 | 41.8 |
| V | 20.1 | 23.0 | 24.8 |

(a) **Dataset size**

| Init. | Rand. | IN-SSL | IN-SL |
|-------|-------|--------|-------|
| A | 41.6 | 41.8 | 41.8 |
| V | 23.7 | 24.8 | 26.1 |

(b) **Visual backbone init.**

| Method | Audio | Video |
|--------|-------|-------|
| A-MAE/V-MAE (baseline) | 36.4 | 17.4 |
| *MAViL stage-1* | | |
| + Joint AV-MAE | $36.8_{(+0.4)}$ | $17.7_{(+0.3)}$ |
| + Inter contrast | 38.4 | 21.0 |
| + Intra and Inter contrast | $39.0_{(+2.2)}$ | $22.2_{(+4.5)}$ |
| *MAViL stage-2* | | |
| + Student-teacher learning | $41.8_{(+2.8)}$ | $24.8_{(+2.6)}$ |

| Ep. | 15 | 20 | 25 |
|-----|----|----|----|
| A | 41.4 | 41.8 | 41.8 |
| V | 23.5 | 24.8 | 24.7 |

(c) **Pre-training epoch**

| Sz. | ViT-S | ViT-B | ViT-L |
|-----|-------|-------|-------|
| A | 36.2 | 41.8 | 42.2 |
| V | 20.5 | 24.8 | 27.0 |

(d) **Model size**

(e) **Module-wise contributions**

Table 5: **Additional ablations** and **module-wise contributions** (mAP on AS-20K)

**Summary.** The module-wise contribution is summarized in Table 5e. MAViL enhances uni-modal MAE performance by learning from audio *and* video simultaneously, as evidenced by significant increases in audio mAP (36.4→41.8, +5.4) and video mAP (17.4→24.8, +7.4) on AS-20K. It is noteworthy that while audio and video are jointly used in the pre-training phase, the fine-tuning is performed with distinct uni-modal encoders on each modality. Therefore, the improvements in the uni-modal classification tasks showcase SSL pre-training from both modalities can enhance the performance of individual modalities.

## 4.5 Main Results

The full model is the stage-2 MAViL student (ViT-B) trained with Eq.(6) for 3 iterations. We quantitatively compare it with other previous models on the 7 benchmark tasks.

**Audio-Video Event Classification.** Table 6 presents a comparison of MAViL's audio (A), video (V), and audio-video (A+V) fine-tuning performance on AudioSet (AS) and VGGSound, alongside recent baselines. Notably, MAViL achieves new state-of-the-art results in audio-only and audio+video classification tasks across these datasets. On the balanced AS-20K, unbalanced AS-2M, and VGGSound classification tasks, MAViL surpasses CAV-MAE [32] (both models are with ViT-B encoders) in A, V, and A+V tasks by a large margin. This improvement can be attributed to the benefits of reconstructing *aligned* and *contextualized* representations over raw inputs and the enhanced contrastive learning through both *intra*-modal and *inter*-modal contrast. Furthermore, MAViL outperforms data2vec [7], highlighting the advantage of utilizing aligned multimodal contexts (Eq.(5)) over uni-modal contexts as the reconstruction targets.

| Method | PT | AS-20K (mAP↑) | | | AS-2M (mAP↑) | | | VGGSound (Acc.↑) | | |
|---|---|---|---|---|---|---|---|---|---|---|
| | | A | V | A+V | A | V | A+V | A | V | A+V |
| *Audio-only Models* | | | | | | | | | | |
| Aud-SlowFast [44] | - | - | - | - | - | - | - | 50.1 | - | - |
| VGGSound [11] | - | - | - | - | - | - | - | 48.8 | - | - |
| PANNs [47] | - | 27.8 | - | - | 43.9 | - | - | - | - | - |
| AST [30] | IN-SL | 34.7 | - | - | 45.9 | - | - | - | - | - |
| HTS-AT [12] | IN-SL | - | - | - | 47.1 | - | - | - | - | - |
| PaSST [49] | IN-SL | - | - | - | 47.1 | - | - | - | - | - |
| Data2vec [7] | AS-SSL | 34.5 | - | - | - | - | - | - | - | - |
| SS-AST [31] | AS-SSL | 31.0 | - | - | - | - | - | - | - | - |
| MAE-AST [5] | AS-SSL | 30.6 | - | - | - | - | - | - | - | - |
| Aud-MAE [41] | AS-SSL | 37.0 | - | - | 47.3 | - | - | - | - | - |
| *Audio-Video Models* | | | | | | | | | | |
| G-Blend [88] | - | 29.1 | 22.1 | 37.8 | 32.4 | 18.8 | 41.8 | - | - | - |
| Perceiver [42] | - | - | - | - | 38.4 | 25.8 | 44.2 | - | - | - |
| Attn AV [24] | IN-SL | - | - | - | 38.4 | 25.7 | 44.2 | - | - | - |
| CAV-MAE [32] | IN-SSL, AS-SSL | 37.7 | 19.8 | 42.0 | 46.6 | 26.2 | 51.2 | 59.5 | 47.0 | 65.5 |
| MBT*[64] | IN21K-SL | 31.3 | 27.7 | 43.9 | 41.5 | 31.3 | 49.6 | 52.3 | **51.2** | 64.1 |
| MAViL | AS-SSL | 41.6 | 23.7 | 44.6 | **48.7** | 28.3 | 51.9 | 60.6 | 50.0 | 66.5 |
| MAViL | IN-SSL, AS-SSL | **41.8** | **24.8** | **44.9** | **48.7** | **30.3** | **53.3** | **60.8** | 50.9 | **67.1** |

Table 6: **Comparison to prior work on AudioSet (AS-20K, AS-2M) and VGGSound** in the audio (A), video (V) and audio+video (A+V) classification tasks. PT: pre-training dataset and type; IN: ImageNet; SL: supervised learning; SSL: self-supervised learning; *:We de-emphasize the model using non-standard dataset splits. We bold the best-performing single model.

The fully self-supervised MAViL also demonstrates superior performance to supervised audio-video models such as MBT [64] in A and A+V classification tasks on AS-20K, AS-2M, and VGGSound. For video classification on AudioSet, it is worth noting that the fully self-supervised trained video backbone still lags behind MBT which pre-trained with the supervision from ImageNet-21K (11 times larger than ImageNet). This disparity is a consequence of the presence of noise and irrelevant visual context in AudioSet as also discussed in [61]. Such bias could make the visual pre-training sub-optimal on AudioSet. We consider resolving this dataset limitation as the future work.

**Transfer to Audio/Speech-only Tasks.** To access the generalizability of the learned audio representations, we further evaluate the AS-pre-trained MAViL by transferring it to other out-of-domain speech-only or audio-only tasks. Specifically, we conduct experiments on the Environmental Sound Classification (ESC-50) [72] and Speech Commands (SPC-v1) [89], where only the audio branch of MAViL is fine-tuned. The results, presented in Table 7, demonstrate that MAViL outperforms recent supervised and self-supervised models, establishing a new state-of-the-art performance on these benchmarks. These findings indicate the desirable transferability of MAViL from audio-video self-supervised pre-training to audio-only downstream tasks.

| Method | PT | ESC-50 | SPC-1 |
|---|---|---|---|
| AST [30] | IN-SL | 88.7 | 95.5 |
| SS-AST [31] | AS-SSL | 88.8 | 96.0 |
| Aud-MAE [41] | AS-SSL | 94.1 | 96.9 |
| MAViL | AS-SSL | 94.4 | 97.3 |
| MAViL | IN-SSL, AS-SSL | **94.4** | **97.4** |

Table 7: **Audio-only tasks** (Acc.↑)

| Method | PT data | MSR-VTT | YouCook |
|---|---|---|---|
| AVLNet [79] | HT100M | 20.1 | 30.7 |
| TVLT [83] | HT100M | 22.6 | 31.8 |
| MAViL | AS-2M | 22.8 | 32.2 |
| MAViL | HT-100M | **23.8** | **33.1** |

Table 8: **Audio-to-video retrieval** (R@1↑)

**Audio-to-Video Retrieval.** MAViL learns aligned audio-video representations that are suitable for textless cross-modal retrieval. To verify this, we further conduct audio-to-video retrieval experiments on MSR-VTT [96] and YouCook [96]. In these tasks, the audio track of a video serves as the query, and the model performs a search over the testing video collection by computing and ranking the similarity between the query embedding and the video embeddings. We fine-tune MAViL using the audio-video pairs in the training sets with Eq.(6). We report recall@1 on the testing sets.

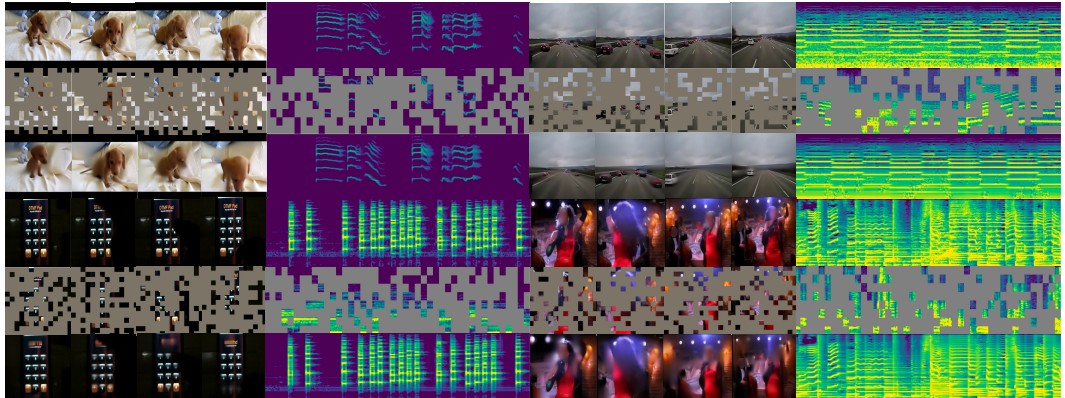

Figure 3: **Video clip and spectrogram reconstruction on the AudioSet *eval* set**. We sample 4 paired (video, audio) examples as follows (click the links to play): Top left: a puppy video; Top right: a recording from an ambulance's dash camera; Bottom left: a person dialing a phone in a dark room; Bottom right: a singer dancing. Input masking ratio: 70%. In each 3-row group, we show the original video and its audio spectrogram (top), masked input to MAViL (middle), and MAViL's video and audio spectrogram reconstructions (bottom). The spectrogram shape is $1024 \times 128$; patch size is $16 \times 16$. Each spectrogram has $64 \times 8 = 512$ patches. After applying 70% masking, there are 154 patches visible to MAViL. The 8-frame (4-second under 2 fps) video clip size is $8 \times 3 \times 224 \times 224$; patch size is $16 \times 16$. Each video has $4 \times 14 \times 14 = 784$ patches after patch embedding (temporal kernel/stride=2). After applying 70% masking, there are 235 patches visible to MAViL.

Following AVLNet [79] and TVLT [83], we explore an alternative setup where we pre-train MAViL on HowTo100M [59], a dataset consisting of 1.3 million instructional videos, instead of using AudioSet for pre-training. It's worth noting that there are notable domain differences between the pre-training datasets (AudioSet or HowTo100M) and the downstream datasets (MSR-VTT or YouCook). Table 8 demonstrates that MAViL surpasses supervised pre-trained AVLNet and self-supervised pre-trained TVLT, achieving new state-of-the-art performance on these tasks.

**Raw Audio-Video Reconstructions** In Fig. 3, we employ a stage-1 MAViL (ViT-B) to reconstruct raw audio spectrograms and video frames with masked inputs. The model is trained using an 80% masking ratio on the AudioSet-2M full training set with *un-normalized* raw spectrograms and video frames as the reconstruction targets (Eq.(4), stage-1). We visualize the reconstruction results by MAViL's audio and video decoders, wherein 70% of the input tokens are masked to its encoders. This visualization is performed on the AudioSet *eval* set.

The results show that MAViL accurately reconstructs the highly corrupted versions of both audio spectrograms and video frames. The reconstructions for videos exhibit high fidelity and preserve spatial and temporal consistency of visual objects (*e.g*., the nearby moving cars recorded by the ambulance's dash camera). For audio reconstructions, MAViL accurately maintains the positions and arrangements of time-frequency components in the spectrogram (*e.g*., the ambulance's siren and the song by the singer). Furthermore, the reconstructed audio and video components are consistent and well-aligned in time, enhancing the overall coherence of the reconstructed content.

## 5   Conclusion

We have presented MAViL, a self-supervised audio-video representation learning framework where masked autoencoding meets contrastive learning. By leveraging an encoder-fusion-decoder architecture, MAViL effectively utilizes complementary information from all modalities for masked autoencoding. It facilitates efficient contrastive learning in both inter-modal and intra-modal scenarios, even with a high 80% masking ratio. Furthermore, MAViL highlights a novel pre-training task with self-training to predict homogeneous contextualized audio-video representations. This approach outperforms conventional uni-modal and multimodal MAEs that predict heterogeneous raw inputs. As a result, MAViL achieves state-of-the-art performance across 7 audio-video classification and retrieval tasks, as well as audio-only tasks under the scalable self-supervised learning setup.

**Acknowledgements.** We thank Luke Zettlemoyer, Kaiming He, Juncheng (Billy) Li, and Florian Metze for their feedback and discussions.

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

# Appendix

This appendix are organized as follows: In §A, we offer the comprehensive experimental details and hyperparameter configurations for pre-training and fine-tuning on each dataset. In §B, we perform additional experiments to evaluate and analyze MAViL's performance. These experiments include:

1. Modality-wise masking ratio and masking type analysis.
2. Contrastive weights/ hyper-parameters analysis.
3. From-scratch and large model analysis.
4. Linear probing analysis.
5. Text-audio retrieval tasks on AudioCaps [45] and Clotho [21].

In §C, we discuss MAViL's societal impact and limitations.

## A    Experimental Details & Hyper-parameters

In this section, we provide additional experimental details for data preprocessing, implementation, pre-training, fine-tuning, and inference. The hyper-parameters are summarized in Table 9. The codebase and the pre-trained models will be available.

### A.1    Data Preprocessing

In our study, we obtained a total of 2.01 million AudioSet videos, including both the video and audio tracks from the balanced and unbalanced training set and the evaluation set. Additionally, we managed to collect 198K VGGSound videos. As part of the preprocessing, we resized the video tracks to 360p while maintaining the aspect ratio and adjusting the longer dimension to 360 pixels. We also resampled the audio tracks to a sampling rate of 16K. We employed different temporal footprints for modeling the audio and video in MAViL, specified as the following:

Following the preprocessing in [64, 30, 41], we transform a raw audio (with mono-channel and under 16K sampling rate) into 128 Mel-frequency bands used in Kaldi [74]. This transformation involves using a 25ms Hanning window that shifted every 10ms. We then normalize the spectrogram according to the mean and variance in each dataset. For a 10-second audio, the resulting spectrogram has a dimension of $1024 \times 128$.

Regarding the video part, we utilize 4-second clips consisting of 8 frames captured at a rate of 2 frames per second (fps). Each input frame has a size of $224 \times 224$. In the pre-training phase, we apply common data augmentations such as random horizontal flip (with a probability of 0.5) and multi-scale random cropping (with a scale ranging from 0.2 to 1.0). In contrast, we apply only center cropping during the testing or inference phase. When processing a 10-second video clip from AudioSet, we randomly sample a starting point and extracted the consecutive 4 seconds of the video (cyclically looping back to the beginning if it was shorter than 4 seconds). As a result, the video clip input, consisting of 3 channels, had dimensions of $8 \times 3 \times 224 \times 224$.

### A.2    Implementation

**Uni-modal Encoders.** We adopt the main design choices from original MAE for images [37] and Audio-MAE [41]. Specifically, we employ separate 12-layer Transformers with 12 attention heads as the encoders for each modality. The patch embedding and positional embeddings layers are also separated for each modality. During our investigation, we explored alternative designs, including sharing the audio-video encoder weights with separated inputs or concatenating them as done in Multi-MAE [6]. However, these alternative architectures resulted in inferior performance compared to the proposed architecture of using separated encoders with separated inputs. As a result, we chose to adhere to the original design of separate encoders for each modality.

In all Transformer encoders (with ViT-B as the default), the embedding dimension $H$ is set to 768 For each input spectrogram of size $1024 \times 128$ representing a 10-second audio, we tokenize it into non-overlapping $16 \times 16$ spectrogram patches using an audio patch embedding layer. The kernel and stride sizes for both the time and frequency dimensions are 16, resulting in a total of $64 \times 8$

spectrogram patches or tokens for the audio sequence. The flattened audio token sequence has a length $N$ of 512. Each audio token corresponds to a 768-dimensional vector. After appending the [CLS] token, adding positional embeddings, and applying 80% masking, the final input audio token sequence is represented as $\mathbf{a}' \in \mathbb{R}^{102 \times 768}$.

For each video clip with dimensions $8 \times 3 \times 224 \times 224$ (4 seconds in duration), we tokenize it into non-overlapping cells using a video patch embedding layer. The spatial kernel and stride sizes are set to 16, while the temporal kernel and stride sizes are set to 2. This process results in a total of $4 \times 14 \times 14 = 784$ video patches or tokens. The flattened video token sequence has a length $M$ of 784. Each video token corresponds to a 768-dimensional vector. After appending the [CLS] token, adding positional embeddings, and applying 80% masking, the final input video token sequence is represented as $\mathbf{v}' \in \mathbb{R}^{156 \times 768}$.

**Fusion Encoders.** Following the ViT-B uni-modal encoders, we incorporate an audio-video *fusion* encoder. The fusion encoder consists of a two-layer (with $L$=2) Transformer, which can be either a vanilla Transformer or an MBT Transformer [64].

In the vanilla Transformer setup, the fusion encoder, denoted as $g_{\text{av}}(\cdot)$, jointly encodes the audio and video tokens. This is done by concatenating the output of the uni-modal encoders for audio $(\mathbf{aum}^{l+1})$ and video $(\mathbf{vum}^{l+1})$ as input, resulting in $(\mathbf{a}_{\text{um}}^{l+1} \| \mathbf{v}_{\text{um}}^{l+1}) = \text{Transformer}^l(\mathbf{a}_{\text{um}}^l \| \mathbf{v}_{\text{um}}^l)$, where "$\|$" denotes concatenation.

In the MBT setup, we extend the vanilla Transformer by appending an additional 4 trainable MBT tokens for each modality. MBT encourages the model to more selectively collate and condense relevant information in each modality by forcing information exchange between modalities to pass through a small number of learnable bottleneck features $\mathbf{b}^0 = [b_1 \ldots b_4], b_i \in \mathbb{R}^H$. The use of MBT tokens was originally proposed in the context of supervised audio-video learning. Precisely, $\mathbf{a}_{\text{um}}^{l+1} \| \mathbf{b}_{\text{a}}^{l+1} = g_{\text{av}}^l(\mathbf{a}_{\text{um}}^l \| \mathbf{b}^l)$ and $\mathbf{v}_{\text{um}}^{l+1} \| \mathbf{b}_{\text{v}}^{l+1} = g_{\text{av}}^l(\mathbf{v}_{\text{um}}^l \| \mathbf{b}^l)$, where $\mathbf{b}^{l+1} = (\mathbf{b}_{\text{a}}^{l+1} + \mathbf{b}_{\text{v}}^{l+1})/2$.

**Decoders.** The audio and video decoders are 8-layer Transformers with an embedding dimension of 512 and 16 attention heads. In the top decoder layer, we applied a linear prediction head to either predict the raw audio spectrogram and video frame patches in stage-1 (*i.e.*, $\mathbf{a}^{\text{raw}} \in \mathbb{R}^{H_{\text{raw}}^{\text{a}}}$ and $\mathbf{v}^{\text{raw}} \in \mathbb{R}^{H_{\text{raw}}^{\text{v}}}$), or predict the aliened and contextualized representations in stage-2 (*i.e.* $\mathbf{a}^{\text{Teacher}}, \mathbf{v}^{\text{Teacher}}, \tilde{\mathbf{a}}, \tilde{\mathbf{v}} \in \mathbb{R}^H$). The audio/video encoder and decoder in MAViL have 86M and 27M parameters, respectively. The floating point operations (FLOPs) for the audio encoder are 48.6G, comparable to the audio encoders in Audio-MAE [41] and CAV-MAE [32].

### A.3 Training and Inference

**Pre-training.** MAViL operates under a fully self-supervised learning setup for pre-training. For pre-training MAViL's audio branch, we randomly initialize it from scratch. For the visual branch, we either randomly initialize it or initialize it with the self-supervised MAE [37] pre-trained on ImageNet where we simply repeat and inflate the convolution kernel in its patch-embedding to handle the additional temporal domain. Different visual initialization methods are compared in Table 6 in the main paper and Table 15 in Appendix. Importantly, MAViL operates under the fully *self-supervised* setup.

MAViL is pre-trained on the combined unbalanced and balanced training sets of AS-2M. The pre-training process is performed using 64 GPUs with a 512 accumulated batch size. In stage-1 and each iteration of stage-2 (for $K = 3$ iterations), we pre-train the model for 20 epochs. Each pre-training session takes approximately 20 hours to complete. In total, the pre-training process takes around 80 hours. Note that the effective learning rate ($lr_{\text{eff}}$) depends on the base learning rate ($lr_{\text{base}}$) and the batch size. Precisely, $lr_{\text{eff}} = lr_{\text{base}} * \frac{\text{batch size}}{256}$. In our experiments, we also tried using strong data augmentations (*e.g.*, mixup [99], SpecAug [99], and CutMix [97]) to augment audio spectrograms during the pre-training phase. However, we observed that the resulting performance was either similar or worse compared to the baseline. Therefore, by default, we exclude these strong data augmentations for both audio and video during the pre-training phase.

Besides the proposed two-stage training framework, we also experimented with a one-stage variant that exploits the exponential move average (EMA) of MAViL (with raw input as the targets) as the teacher (similar to data2vec [7]) to generate contextualized targets to be reconstructed by a separated decoder head in each modality. This setup results in less stable training behavior in the

| Configuration | Pre-training AS-2M PT | Fine-tuning AS-2M | AS-20K | VGGSound | ESC | SPC |
|---|---|---|---|---|---|---|
| Optimizer | | AdamW [56] | | | | |
| Optimizer momentum | | $\beta_1 = 0.9, \beta_2 = 0.95$ | | | | |
| Weight decay | | 0.00001 | | | | |
| Base learning rate | 0.0002 | $0.0001^{\dagger}$ | 0.001 | 0.0002 | 0.0005 | 0.001 |
| Learning rate schedule | | half-cycle cosine decay [55] | | | | |
| Minimum learning rate | | 0.000001 | | | | |
| Gradient clipping | | None | | | | |
| Warm-up epochs | 4 | 20 | 4 | 4 | 4 | 1 |
| Epochs | 20 | 100 | 60 | 60 | 60 | 10 |
| Batch size | 512 | 512 | 64 | 256 | 64 | 256 |
| GPUs | 64 | 64 | 8 | 32 | 4 | 4 |
| Weighted sampling | False | True | False | True | False | False* |
| Weighted sampling size | - | 200,000 | - | 200,000 | - | - |
| Augmentation | R | R | R | R+N | R | R+N |
| SpecAug [68] (time/frequency) | - | 192/48 | 192/48 | 192/48 | 96/24 | 48/48 |
| Drop path [50] | 0.0 | 0.1 | 0.1 | 0.1 | 0.1 | 0.1 |
| Mixup [99] | 0.0 | 0.5 | 0.5 | 0.5 | 0.0 | 0.5 |
| Multilabel | n/a | True | True | False | False | False |
| Loss Function | MSE | BCE | BCE | BCE | CE | BCE |
| Dataset Mean for Normalization | -4.268 | -4.268 | -4.268 | -5.189 | -6.627 | -6.702 |
| Dataset Std for Normalization | 4.569 | 4.569 | 4.569 | 3.260 | 5.359 | 5.448 |

Table 9: **Pre-training (PT) and Fine-tuning (FT) hyper-parameters**. For augmentation, R: sampling random starting points with cyclic rolling in time; N: adding random noise (signal-to-noise ratio (SNR): 20dB) to spectrograms. For loss functions, BCE: binary cross entropy loss (for multi-label datasets or when using mixup); CE: cross-entropy loss, MSE: mean square error loss. *: We repeat and balance each class to 50% of the size of the unknown class. $^{\dagger}$: For ViT-S, We use a learning rate of 0.0005 on AS-2M FT and 0.002 on AS-20K FT for the ViT-S model. For the ViT-L model, we use 0.0001 and 0.0005 for AS-2M and AS-20K FT experiments.

audio-video scenario. The instability is presumably a consequence of different convergence patterns in each modality, which is also observed in [88]. Consequently, this makes the design and tuning of mixture/update scheduler sub-optimal (i.e., that controls the EMA update in Data2vec Sec 3.3). Regrettably, this led to notable performance degradation:

| | 2-stage | 1-stage (EMA) |
|---|---|---|
| Audio | 41.8 | 40.3 |
| Video | 24.8 | 21.7 |

Table 10: **2-stage vs. 1-stage (EMA)**

By employing a 2-stage training process, MAViL not only simplifies the methodology and achieves better performance, but it also mitigates the necessity of devising intricate update schedules and carefully tuned EMA hyperparameters, as were required in prior work.

**Fine-tuning.** We fine-tune MAViL in three scenarios: (1) audio-only, (2) video-only, and (3) audio+video. We follow the setup in MAE and retain only the pre-trained uni-modal encoders for fine-tuning. In the audio-only and video-only setups, we fine-tune the respective encoders in the MAViL (stage-2). In the audio+video fusion setup, we introduce a 2-layer vanilla Transformer on top of the audio and video encoder in the MAViL (stage-2) and fine-tune it using both audio and video inputs. The hyperparameter configurations specified in Table 9 are employed for finetuning on each dataset. Empirically we observed a discrepancy in convergence rate between audio and video. We circumvent this by applying a 50% learning rate reduction for the weights of the video encoder when performing audio+video fusion fine-tuning.

We adopt the standard fine-tuning pipeline and augmentation in prior audio/audio-video classification works [30, 41, 64]. Specifically, we employ SpecAug [68], mixup [99], balanced sampling [51], and

fine-tuning masking [41] (a 20% random masking rate for time and frequency in audio spectrograms; 20% for space and time in video clips). For video, we use standard video augmentations used in video classification [90, 27].

To perform importance sampling that balance the fine-tuning scheme on the unbalanced AS-2M (and VGGSound), we apply a distributed weighted sampler as prior works [51, 30, 12, 49]. We set the probability of sampling a sample proportional to the inverse frequency of its labels, where the label frequency is estimated over the training set. Specifically, for a instance $i$ in a dataset $\mathcal{D}$ with a label pool $\mathbf{C}$, its sampling weight is proportional to $\sum_{c_i \in \mathbf{C}} w_c$, where $w_c = \frac{1000}{\sum_{i \in \mathbf{D}} c_i + \epsilon}$ and $\epsilon = 0.01$ is set to avoid underflow in majority classes. During the fine-tuning process on AS-2M, we randomly sample 200K instances (approximately 10% of AS-2M) with replacement in each epoch. We fine-tune MAViL for 100 epochs, which corresponds to approximately 10 full epochs of AS-2M. The entire fine-tuning process typically takes around 10 hours to complete.

**Inference.** After fine-tuning, we select the last checkpoint for inference. For the video and audio+video tasks, we adopt the standard approach used in video action recognition [26, 23, 52] by uniformly sampling ten 4-second video clips throughout the time domain of a video. Each of these sampled video clips is individually fed forward through the model to generate predictions. Note that for audio+video classification, the audio input remains the same 10-second audio recording throughout the sampling of video clips.

| # Clips (AS-2M) | 1 | 10 |
|---|---|---|
| Audio | 48.7 | 48.7 |
| Video | 29.4 | 30.3 |
| Audio+Video | 52.6 | 53.3 |

Table 11: **Number of video clips in the inference time.**

We average the ten predictions as the instance-level prediction and report the classification performance in Table 6 in §4. Note that these results are based on single-modal predictions, without ensembling multiple models. In Table 11, we compare the results obtained from one-clip predictions and ten-clip predictions (mAP on AS-2M). The sampling of ten clips leads to improvements of up to 0.9 mAP for video-only and audio+video tasks, while the audio-only task remains unaffected.

# B   Additional Experiments and Analysis

In this section, we present additional analysis to extend the study of the module-wise contribution in Table 12. We then expand our study on another important type of audio task: text-audio retrieval.

We organize this section as follows: Firstly, we investigate how different choices of masking ratio and masking type may affect the model performance. Next, we examine the effects of adjusting contrastive weights in the training objective. By exploring different weight settings, we aim to understand the influence of contrastive learning on the model's ability to capture audio-video relationships. Furthermore, we compare different approaches to visual backbone initialization and evaluate the performance using larger (ViT-L) audio/video encoders in MAViL-Large models. This analysis helps us understand the benefits and trade-offs of using larger backbone models and different initialization strategies. Additionally, besides audio-video classification tasks and audio-video retrieval tasks presented in the main paper. We include our study on audio-text retrieval tasks in the last.

| Method | Audio | Video |
|---|---|---|
| A-MAE/V-MAE (baseline) | 36.4 | 17.4 |
| *MAViL stage-1* | | |
| + Joint AV-MAE | $36.8_{(+0.4)}$ | $17.7_{(+0.3)}$ |
| + Intra and Inter contrast | $39.0_{(+2.2)}$ | $22.2_{(+4.5)}$ |
| *MAViL stage-2* | | |
| + Student-teacher learning | $41.8_{(+2.8)}$ | $24.8_{(+2.6)}$ |

Table 12: **Module-wise Contribution** in MAViL).

## B.1 Masking Ratio and Type

In addition to applying a shared masking ratio for each modality, we also investigated the impact of applying different masking ratios for audio and video. The results of this analysis are summarized in Table 13a. Interestingly, we did not observe a significant change in performance (mAP on AS-20K) when using different masking ratios for audio and video. Based on these findings, we simplify the approach by defaulting to an 80% masking ratio for both audio and video, as the Joint AV-MAE entry (the second row) in Table 12.

| Ratio | 70% (A) | 80% (A) | 90% (A) |
|---|---|---|---|
| 70% (V) | 36.7/17.5 | 36.8/17.5 | 36.4/17.3 |
| 80% (V) | 36.7/17.2 | 36.8/17.7 | 36.8/17.4 |
| 90% (V) | 36.5/17.3 | 36.6/17.6 | 36.8/17.5 |

(a) **Modality-wise Masking**

| Type | 70% | 80% | 90% |
|---|---|---|---|
| Random (A), Random (V) | 36.7/17.5 | 36.8/17.7 | 36.8/17.5 |
| Time-Freq (A), Random (V) | 36.2/17.5 | 36.3/17.7 | 36.3/17.8 |
| Random (A), Space-Time (V) | 36.7/17.2 | 36.7/17.3 | 36.8/17.5 |
| Time-Freq (A), Space-Time (V) | 36.0/17.1 | 36.2/17.1 | 36.3/17.3 |

(b) **Masking Type**

Table 13: **Masking Ratio** and **Masking Type** (mAP on AS-20K).

The default masking strategy in our model is random masking, which applies the same Bernoulli trial parameterized by a masking ratio ($p$) to each spectrogram or RGB patch. In Table 13b, we explored more advanced masking strategies and compare their impacts. For audio spectrogram, in addition to random masking (time-and-frequency agnostic with Bernoulli trials), we investigated time-masking (randomly masks multiple periods of time components) and frequency masking (randomly masks multiple frequency bands). We perform Bernoulli trials on time or frequency slots instead of individual patches. For video frames, we explored time-wise masking (randomly masking an entire frame) and space-wise masking (randomly masking a spatial patch across time). We set the masking ratio between spatial/frequency and time as 2:1 and adjusted the overall ratio from 70% to 90% for comparison with random masking.

Surprisingly, we do not observe improvements when applying these advanced masking strategies for multimodal pre-training. The simplest random masking approach achieved the best pre-training performance. This observation aligns with the findings in uni-modal MAEs [37, 27, 41], suggesting that the random masking strategy is effective and sufficient for multimodal pre-training.

## B.2 Contrastive Weights

Table 14 showcases the impact of adjusting contrastive weights $\alpha$ and $\beta$ in MAViL. The results show that fine-tuning these contrastive weights leads to improved performance. In our experiments, we set $\alpha = 0.1$ and $\beta = 0.01$ which yield the best performance.

It is important to note that the smaller contrastive weights in Eq.(4) do not imply that the contrastive objectives are less significant. The weights are chosen to scale and balance the gradients from the reconstruction and the two contrastive objectives to ensure they fall within a comparable range. This adjustment enhances training stability. Furthermore, the softmax temperatures used in NCE (Eq. (2)) are set as $\tau_c^{inter} = 0.1$ (more tolerant) for inter-modal contrastive learning and $\tau_c^{intra} = 1.0$ (stricter) for intra-modal contrastive learning. These temperature values help regulate convergence across modalities in the contrastive learning process.

| $\alpha$ | 0.3 | 0.1 | 0.05 |
|---|---|---|---|
| Audio | 41.5 | 41.8 | 41.4 |
| Video | 24.3 | 24.8 | 24.4 |

(a) **Inter-modal** $\alpha$

| $\beta$ | 0.1 | 0.05 | 0.01 |
|---|---|---|---|
| Audio | 41.3 | 41.5 | 41.8 |
| Video | 24.3 | 24.7 | 24.8 |

(b) **Intra-modal** $\beta$

Table 14: **Contrastive Weights** (mAP on AS-20K).

## B.3 From-scratch Visual Backbone and Large Models

Under the fully self-supervised setup, MAViL initializes its audio branch from scratch and initialize its visual branch either from scratch or from a ImageNet self-supervised pre-trained MAE (IN-SSL). In

| | | | AS-20K | | | AS-2M | | |
|---|---|---|---|---|---|---|---|---|
| Model | A-init | V-init | A | V | A+V | A | V | A+V |
| MAViL-Base | scratch | IN-SSL | 41.8 | 24.8 | 44.9 | 48.7 | 30.3 | 53.3 |
| MAViL-Base | scratch | scratch | 41.6 | 23.7 | 44.6 | 48.7 | 28.3 | 51.9 |
| MAViL-Large | scratch | IN-SSL | 42.1 | 27.1 | 45.3 | 48.8 | 32.4 | 53.3 |
| MAViL-Large | scratch | scratch | 42.3 | 25.3 | 45.1 | 49.1 | 30.6 | 52.5 |

Table 15: **Visual Backbone Initialization and Model Size** (mAP).

this part, we further explore and compare the visual backbone initialization strategies under different model sizes.

As shown in the top two rows of Table 15, when considering MAViL-Base models, there is a small gap (-0.2 mAP on AS-20K) observed in the audio stream when discarding visual initialization from the ImageNet self-supervised model. However, a larger gap (-0.9 mAP) is observed in the video stream. A similar trend is observed in the AS-2M experiments. This discrepancy in the visual part can likely be attributed to biases and visual quality issues such as misalignment, title-only content, and low-resolution videos present in AudioSet.

To address this gap in the visual part, incorporating additional uni-modal pre-training steps could potentially improve model performance. For instance, conducting separate audio-only and video-only large-scale pre-training as the first step. In this work, we focus on audio-video pre-training solely on AudioSet for simplicity and for fair comparison with baselines. The possibility of incorporating additional pre-training steps is left for future research.

When using large models (ViT-L, rows 3-4), the gap in visual mAP (-1.8 mAP) still persists. Interestingly, the audio part of large models actually benefits from from-scratch visual initialization, showing an improvement of +0.2-0.3 mAP. Additionally, when comparing rows 1-2 to rows 2-3, the visual stream is benefited more by employing a larger (ViT-L) backbone. Across all the configurations (from-scratch or visual initialization with IN-SSL), MAViL consistently outperforms recent baselines (in Table 6 of the main paper) by a significant margin.

## B.4    Linear Probing Results

Additionally, we evaluate MAViL's representations by performing linear probing on ESC and AudioSet-20K. As shown in Table 16, MAViL's representations outperform other recent baselines by a large margin. There is still is a gap between linear-probing and end-to-end fine-tuning.

| | PT-Data | ESC-FT | ESC-Linear | AS-20K-FT | AS-20K-Linear |
|---|---|---|---|---|---|
| XDC | A+V,AS | - | 84.8 | - | - |
| AVID | A+V,AS | - | 89.1 | - | - |
| BraVe | A+V,AS | - | 90.4 | - | - |
| CrissCross | A+V,AS | - | 90.5 | - | - |
| SS-AST | A,AS | 88.8 | 85.6 | 31.0 | - |
| A-MAE | A,AS | 94.1 | 89.5 | 37.0 | - |
| MAViL | A+V,AS | 94.4 | 90.8 | 41.8 | 30.0 |

Table 16: **Linear-probing and fine-tuning on ESC and AS-20K**. PT: Pre-training. FT: end-to-end fine-tuning. Linear: Linear-probing under standard protocol. AS: AudioSet. Metric: Top-1 accuracy for ESC-50 and mAP for AS-20K.

## B.5    Text-Audio Tasks

Another important audio-centered multimodal application involves text-to-audio and audio-to-text retrieval tasks. In text-to-audio retrieval, the query is a text description, and the model performs a search over the (testing) audio collection by computing and ranking the similarity between the query embedding and the audio embeddings. To evaluate the audio representations learned by MAViL, following CLAP [94], we add a text encoder initialized from Roberta [54]. We perform fine-tuning

with inter-modal contrast on the same training set used by CLAP. Specifically, AudioCaps [45] and Clotho [21], and LAION-630K [94]. In Table 17, we report recall@1, 5, and 10 on the testing sets.

| | AudioCaps [45] | | | | | | Clotho [21] | | | | | |
| | Text-to-Audio | | | Audio-to-Text | | | Text-to-Audio | | | Audio-to-Text | | |
| Model | R@1 | R@5 | R@10 | R@1 | R@5 | R@10 | R@1 | R@5 | R@10 | R@1 | R@5 | R@10 |
|---|---|---|---|---|---|---|---|---|---|---|---|---|
| MMT* [66] | 36.1 | 72.0 | 84.5 | 39.6 | 76.8 | 86.7 | 6.7 | 21.6 | 33.2 | 7.0 | 22.7 | 34.6 |
| ML-ACT* [58] | 33.9 | 69.7 | 82.6 | 39.4 | 72.0 | 83.9 | 14.4 | 36.6 | 49.9 | 16.2 | 37.6 | 50.2 |
| CLAP [94] | 32.7 | 68.0 | 81.2 | 43.9 | 77.7 | 87.6 | 15.6 | 38.6 | 52.3 | 23.7 | 48.9 | 59.9 |
| MAViL | **37.3** | **72.8** | **84.5** | **49.3** | **81.8** | **91.5** | **17.2** | **41.0** | **53.5** | **23.3** | **49.5** | **63.6** |

Table 17: **Text-to-Audio retrieval** and **Audio-to-Text retrieval** (R@1,5,10↑) on AudioCaps and Clotho. *: models trained without LAION-630K [94].

As shown above, MAViL significantly outperforms CLAP and other recent audio-text models, achieving new state-of-the-art performance on both audio-to-text and text-to-audio retrieval tasks. These results further validate the effectiveness of MAViL's representations not only in audio-video and audio-only tasks, but also in audio-text tasks.

## C   Limitations and Impacts

**Limitations.** There are several limitations associated with MAViL. Firstly, the scale of the data poses a limitation. The AudioSet [28] dataset used by MAViL, with two million samples, is approximately two orders of magnitude smaller than the text corpora used in recent language models [20, 54, 9]. It is also an order smaller than image corpora like ImageNet-21K used by MBT [64].

Another limitation pertains to the duration of each audio sample. The 10-second recording in AudioSet are relatively short, which can hinder the proper learning of distant temporal dependencies in audio and video. This limitation restricts the potential applicability of MAViL to tasks that require modeling longer audio sequences, such as automatic speech recognition (ASR). Regarding video modeling, due to GPU memory constraints and choice of video footprints, MAViL only models 4-second video segments. This limitation makes it challenging to effectively model long video sequences. Additionally, the presence of low-quality videos and misaligned audio-video pairs in AudioSet may adversely affects pre-training.

**Potential Societal Impacts.** The datasets used in this paper, including AudioSet and other end task datasets, were properly licensed and publicly available at the time of data collection. It is important to note that some of the data may have been removed by YouTube or the dataset uploaders. Most of the data in these datasets are licensed under the Creative Commons BY-NC-ND 4.0 license or the Creative Commons 3.0 International License.

To investigate the bias in AudioSet, we selected 200 videos containing speech. In these videos, we did not observe any visual bias in the sampled speakers, which encompassed a wide range of ages, races, and genders. However, it is possible that there may be biases in the distribution of population and ethnicity within AudioSet. It is important to exercise caution and be aware of the potential unintended gender, racial, and societal biases present in AudioSet, which serves as the pre-training data for MAViL.

Given that AudioSet consists of a vast collection YouTube videos, there is a potential risk that MAViL could learn to reconstruct sensitive personal information, which could then be exploited for malicious purposes, including the creation of audio deepfakes [81, 16]. To address this concern, the released MAViL would be discriminative models, specifically the audio and video encoders, rather than generative models such as decoders. This shift aims to mitigate the potential risks associated with generating synthetic content that could be misused.

