# OpenReview forum: "MAViL: Masked Audio-Video Learners"
_NeurIPS.cc/2023/Conference — NeurIPS 2023 poster_

### Official Review · Reviewer_dLj4 · 2023-06-30

**Soundness:** 3 good
**Presentation:** 3 good
**Contribution:** 2 fair
**Rating:** 4
**Confidence:** 5

**Summary:**

This paper proposes a self-supervised audiovisual representation learning framework. It combines MAE with contrastive learning in both intra and inter modality settings. The authors also suggest a self-training schema where the student model, which only sees masked input, needs to align with the latent representations of the unmasked input produced by the teacher. This shows to improve the quality of learned representations in just a few iterations without relying on external pre-trained teacher models.

**Strengths:**

1. The work is well-motivated and for the most part easy to follow.
2. Experiments and ablation studies are comprehensive with some reservations (ref to Questions section)
3. Components of the proposed approach (Fig 1) are not novel, however putting them together and studying how they interact with one another is valuable.
4. While there are other works which have combined MAE with contrastive learning, this paper has done a good job distinguishing itself from those concurrent/earlier efforts.
5. Self-training results are promising (Table 4.b) and the ablation for reconstruction target (Table 4.a) provides valuable insight.


**Weaknesses:**

1. The effectiveness of the fusion layer is not clear: Table 1 does not show any meaningful gain added by the fusion layer over unimodal MAE. It is worth noting that “None” has fewer parameters compared to other columns in Table 1 due to not having a fusion layer. Similarly in Table 4.a when comparing M-Fusion against M-Uni.
2. Table 3: Intra is lower than Inter, and their combination provides marginal gain over the inter (0.6 on audio and 1.2 on video). It is worth remembering that the addition of intra doubles the computation cost since we need to process two views from each input-modality. Intra loss is one of the main points that this work is using to distinguish itself from CAV-MAE.
3. Table 5.e summarizes the effectiveness of the contributions claimed by MAViL. We can see that the addition of joint AV-MAE adds nothing meaningful to the baseline. Similarly is the marginal gain due to the addition of intra loss over the inter one. The student teacher learning is the only novel contribution that shows to be effective.
4. Minor note (not a weakness): arxiv paper https://arxiv.org/abs/2303.12001  also combines MAE and contrastive losses though the setup and modalities are different from the current paper.

**Questions:**

1. What is difference between $\tilde{a}$ (similarly $\tilde{v}$) in line 183 and $\hat{a}$ (similarly $\hat{v}$) in line 135?
2. Eq 5 is confusing: $\tilde{a}$ is in the raw space (ref Eq 1) but here it is being compared against contextualized latent embedding of the audio (similarly for video). It made sense if $\tilde{a}$ did not have the decoder (i.e $f^{-1}$). The current form also contradicts with default setting in Table 4.a. Authors should clarify.
3. Authors should provide intuition on $\tau$ values for inter and intra losses. Why do we need to be more tolerant for intra? How are the actual values been chosen?
4. Linear probe is more favorable for evaluation than fine-tuning. As also noted in Sec 4.3, there are so many bells and whistles (e.g all sorts of data augmentations and learning rate annealing strategies etc) when it comes to fine-tuning. Linear probe on the other hand relies on minimal parameters and evaluates the quality of learned representations not the pre-trained model. It would be more helpful if we were able to see linear probe results.
5. Table 5.b: Different visual backbone initializations impact the quality of the corresponding modality in a meaningful way (as expected) however the performance  is flat on the audio side. Considering fusion layer is supposed to mix the representations between modalities, one would expect to see a lift for the audio backbone too. It would help if authors provide some insights.
6. Table 6: Is it fair to compare against audio-only models when MAViL was pre-trained with video and audio?

**Limitations:**

Yes, they have.

---

> ### Author Rebuttal · Authors · 2023-08-09
>
> We appreciate your attention to recognizing the unique value of MAViL's multimodal self-training approach, using aligned and contextualized representations with MAE and contrastive learning. Additionally, we thank you for recognizing our comprehensive analysis and experiments. Please see our response to your comments and questions:
>
> **W1 Effectiveness of fusion encoder?**
>
> A-W1: Without fusion, MAViL's performance is expected to drop (\~1 mAP). We acknowledge your observation that audio-video might appear to have a relatively diminished role at first glance, contributing \~0.3\-0.4 mAP over vanilla A/V MAEs. However, as you pointed out, fusion encoder is also significant in stage-2 as it provides contextualized targets. It's important to clarify that, in Table 4a, M-Uni and M-Fusion represent distinct outputs generated by the same teacher model trained with the fusion encoder.
>
> To accurately assess the effectiveness of fusion, we removed fusion layer and kept all other objectives same in stage-1 (Eq 4) and stage-2 (Eq 6). The result of this “leave-fusion-out” experiment is as below:
>
> ||A|V|
> |---|---|---|
> |w/ fusion|41.8|24.8|
> |w/o fusion|40.7 (-1.1)|23.8 (-1.0)|
>
> The consistent drops w/o fusion validate the effectiveness of using the fusion layer for exchanging complementary audio-video context.
>
>
> **W2 Smaller gain of intra-contrast and marginal gain with combination?**
>
> A-W2: We concur with your discovery that inter-modal contrast is more effective (+1.6 (A), +3.3 (V)) than intra-modal contrast (+1.3 (A), +2.1 (V)). It's important to note, however, that these objectives are complementary and mutually reinforcing, as they encourage the alignment of semantically related audio and video representations from distinct angles. By exploiting both inter-contrast and intra-contrast objectives, MAViL attains the best performance (2.2 (A), +4.5 (V)), surpassing the performance of using either objective in isolation.
>
> Also, we acknowledge your comment about additional computation required for encoding the second view. However, unlike conventional contrastive learning methods (e.g., SimCLR, MoCo, Wav2Vec2), MAViL encodes only 20% of tokens. This substantial reduction significantly mitigates the computational burden.
>
>
> **Q1-2 Difference between $\hat{a}$ and $\tilde{a}$? and the space of $\tilde{a}$ in Eq 5?**
>
> A1-2: Apologies for any confusion. $\hat{a}$ and $\hat{v}$ are the prediction of raw spectrograms and raw pixels. In contrast, $\tilde{a}$ and $\tilde{v}$ are the prediction of contextualized audio/video embeddings in the shared latent space.
>
> In stage-2 (Eq 5), MAViL encodes un-masked raw tokens, conducts fusion, and decodes to predict the contextualized embeddings generated by a previously trained teacher’s encoder, encompassing full views. For example, a student's decoder predicts *BxNxH* contextualized audio embeddings (Eq 5) (H: embedding dimension), instead of predicting *BxNx1x16x16* (spec. patches) (Eq 1).
>
> **Q3 Intuition of $\tau$ in inter/intra contrast and the choice? Should intra-modal be stricter/tolerant?**
>
> A3: Thank you for your valuable input. We set temperature $\tau = 0.1$ for inter-modal and \tau=1.0 for intra-modal contrast.  A higher temperature used in the NCE serves to accentuate the differentiation between positive and negative samples, leading to enhanced alignment or distinction. Intuitively, alignment/distinction is more noticeable when comparing samples within the same modality (eg, clips from the same video, versus other videos) compared to samples from different modalities (eg, a scene of a person talking and his speech, vs other possible speeches). With this in mind, we set a stricter temperature $\tau=1.0$ for intra-modal contrast, as the alignment/distinction within the same modality demands more precision. Conversely, for inter-modal we adopt a more permissive \tau=0.1. The best values were selected via a grid search over 0.1, 0.2, 0.5, and 1.0.
>
> **Q4 Linear probe results?**
>
> A4: Thank you for the suggestion. We conducted additional audio probing on ESC and visual probing on HMDB, UCF:
>
> ||PT-Data|ESC50-Linear|
> |---|---|---|
> |XDC|A+V,AS|84.8|-|
> |AVID|A+V,AS|89.1|-|
> |BraVe|A+V,AS|90.4|-|
> |CrissCross|A+V,AS|90.5|-|
> |SSAST|A,AS|85.6|-|
> |A-MAE|A,AS|89.5|-|
> |MAViL|A+V,AS|90.8|30.0|
>
> ||PT-Data|#fms|UCF101-Linear|HMDB51-Linear|
> |---|---|---|---|---|
> |CrissCross|A+V,AS|8|87.7|56.2|
> |MAViL|A+V,AS|8|90.5|58.3|
> |BraVe*|A+V,AS|32*|93.0|69.4|
>
> As shown, MAViL yields the best audio classification and comparable video classification performance (8-frms).
>
> **Q5 Saturated audio mAP with visual backbone initialization?**
>
> A5: This is an insightful observation! MAViL exploits aligned and complementary info present in audio-video pairs. Our approach assumes meaningful connections exist between audios and videos. However, these associations might not be perfect in real-world audio/video. For instance, audios in AudioSet could arise where the corresponding videos possess low visual quality (eg, text-only footages) or irrelevant context. These noises hinder the audio branch from fully capitalizing on the visual branch, which might consequently leads to saturated benefits even with an better visual initialization.
>
> **Q6 Fair to compare with audio-only?**
>
> A6: We believe it's crucial to highlight the value of audio-video pre-training, as it not only amplifies performance in audio-video classification and retrieval but also refines the representations of each individual modality in downstream tasks. Being consistent with other audio-video studies in this field (eg MBT, CAV-MAE, Perceiver, and more), we consider it appropriate to draw comparisons and demonstrate the advantages of audio-video models over audio-only models. This is justified by the fact that these models leverage distinct viewpoints or modalities inherent within the same dataset, rather than incorporating additional data (eg by using the same AudioSet).

---

> > ### Comment · Reviewer_dLj4 · 2023-08-17
> >
> > Authors’ rebuttal has clarified a few aspects of the work. However, I still consider the contribution of fusion layer and intra-loss to be marginal in presence of inter-loss. These are also what the paper relies on in order to distinguish itself from other similar works. Hence, I maintain my original rating.

---

> ### Author Response · Authors · 2023-08-21
> **Thank you for your comment**
>
> Thank you for reviewing our paper, and we are pleased to note that your inquiries have been addressed. It’s important to note that MAViL’s main contribution lies in the advancement of self-supervision via reconstructing aligned and contextualized multimodal embeddings instead of heterogeneous raw signals in each modality. This entire approach is novel and leads to strong performance improvements (+5.4 (A), +7.4 (V) mAP or relatively 15% (A) and 42% (V) on AudioSet to A/V-MAE baselines).
>
> Our ablation experiments validated the distinctive and substantial contribution by each component: fusion-layer (+1.1 (A), +1.0 (V) mAP), intra-loss (+1.3 (A), +2.1 (V) mAP), and notably, the multimodal student-teacher framework (+2.8 (A), +2.6 (V) mAP). With these components working jointly, MAViL sets new state-of-the-arts on 7 popular audio-video benchmarks. We firmly believe these contributions and improvements are not marginal, but hold substantial significance and value for the research community. We would be happy to clarify further if any specifics are unclear. Thank you for your attention and feedback.

---

### Official Review · Reviewer_z1cD · 2023-06-30

**Soundness:** 3 good
**Presentation:** 3 good
**Contribution:** 2 fair
**Rating:** 4
**Confidence:** 5

**Summary:**

The authors present Masked Audio-Video Learners (MAViL), pretrained on 3 pseudo tasks and consisting of a 2-stage approach. i) masked reconstruction (adoption of MAE, AudioMAE, VideoMAE) ii) intra-modal and inter-modal contrastive learning (similar contrastive loss like SIMCLR, MOCO), and iii)  student-teacher learning (performed in stage 2, by minimizing the error between earlier pretrained encoders referred to as teachers and the current encoders referred to as students). The proposed framework is pretrained and evaluated with AudioSet and VGGSound. Additionally, audio-to-video retrieval experiments are done on MSR-VTT and YouCook.




**Strengths:**

- The proposed method is a nice extension of masked reconstruction in a multimodal setup.
- Fair amount of experiments are conducted as part of ablation to support the design choices.
- Good presentation and easy to follow.
- Thanks to the authors for making the code to be public and sharing the details of the implementations which are helpful for reproducibility.

**Weaknesses:**

- This work does not provide results on linear evaluation. The whole point of SSL is to learn rich abstract representations that can be used through minimal training with labelled data, i.e., linear SVM or FC-tuning.
- The comparison setup with prior works is insufficient. For example, the SOTA evaluation on visual tasks is extremely weak and most of the entries in Tab. 6 for the video are blank (-), while there are plenty of methods that can be compared. This work does not compare the results of ESC50 with several AV-SSL prior works.
- The proposed method lacks technical novelty as all the pieces (masked reconstruction, contrastive learning, student-teacher) of the proposed method are already been studied quite heavily in the literature, be it in a uni-modal or multimodal/cross-modal setups.

**Questions:**

1. The method uses 10 seconds of audio and 4 seconds of video, which means the network sees asynchronous audio-video segments. Why is such a setup used? Is there a reason for using different durations of audio-video sequences? How would the results be affected if the same 4 seconds of audio and video are used? Could you please provide such results?

2. What would happen if you drop the fusion encoder and directly reconstruct the encoded representation? Could you please provide the results by removing the fusion encoder while keeping the rest of the setup intact? The loss function would remain the same. To clarify further, replace $\hat{v}$ and $\hat{a}$ in Eq 1 with the following values: $\hat{v} =f^{-1}_v(f_v(v'))$ and $\hat{a} =f^{-1}_a(f_a(a')) $.

3. What would happen if you initialize the encoders with VideoMAE and AudioMAE checkpoints and only perform stage 2, which involves training with Eq. 6? Could you please provide the results for such a scenario? You can utilize the pretraining weights released by the respective papers. This approach would simplify the process, as currently, the intra-modal and inter-modal contrastive losses are optimized twice (in both Eq. 4 and 6), which appears to be somewhat redundant.

4. The authors have maintained a fixed teacher in their approach. However, I'm curious to know how it would perform if the teachers are updated slowly using Exponential Moving Average (EMA). Additionally, the current loss between the teachers and students is Mean Squared Error (MSE). I am curious to understand why MSE is used and what would happen if the loss function is changed to minimize cross-entropy or cosine distance instead. Could you please provide the findings for such experiments?

5. The authors are requested to include the following information in the SOTA comparison tables (Tab. 6) to ensure a fair and informed comparison with other methods:
- model names (e.g., ViT, ResNet, etc.)
- number of parameters during pretraining (backbone+auxiliary network parameters such as heads, decoders, etc.)
- number of params during evaluation,
- pretraining input size
- evaluation input size.

6. The authors are strongly recommended to compare MAViL with prior works in linear evaluation. As a reference for such evaluation, you may see Brave (Tab. 6).

7. The authors are suggested to evaluate their method on the following setups.
- linear and finetune comparison on (i) large-scale datasets like Kinetics400 and Kinetics600; (ii) small-scale datasets like UCF101 and HMDB51; and (iii) temporal-focused datasets like SSv2. Similar to: VideoMOCO, CVRL, p-BYOL, ELO, VideoMAE, Brave, AVID, Robust-xID, Selavi, GDT, CrissCross, STiCA, and CM-ACC.
- action retrieval on UCF101 and HMDB51 to report R@1, R@5, R@20. Similar to: Robust-xID, Selavi, GDT, and CrissCross.
- You may check Brave Tab 6 as a reference for a comprehensive comparison.

8. The authors are also recommended to compare the results of ESC50 with prior  AV-SSL methods, please use similar references for comparison. Refs.: XDC, AVID, Brave, CrissCross, CM-ACC.

9. Typo: in Figure 2: iter 2, V-Enc -> V-Dec for the reconstruction part.

10. Minor suggestions:
- There are a few works that are very similar to MAViL and have some overlapping ideas such as joint-masked reconstruction, student-teacher etc. They are available on arXiv for 5-6 months, I would suggest discussing them as well, e.g., AVMAE and XKD.
- I recommend adding a pseudocode, it gives more clarity.

=========
References:

VideoMAE: https://arxiv.org/pdf/2203.12602.pdf;
Brave: https://arxiv.org/pdf/2103.16559.pdf;
AVID: https://arxiv.org/pdf/2004.12943.pdf;
Robust-xID: https://arxiv.org/pdf/2103.15916.pdf;
Selavi: https://arxiv.org/pdf/2006.13662.pdf;
GDT: https://arxiv.org/pdf/2003.04298.pdf;
CrissCross: https://arxiv.org/pdf/2111.05329.pdf;
CM-ACC: https://arxiv.org/pdf/2009.09805.pdf;
VideoMOCO: https://arxiv.org/pdf/2103.05905.pdf;
CVRL: https://arxiv.org/pdf/2008.03800.pdf;
p-BYOL: https://arxiv.org/pdf/2104.14558.pdf;
ELO: https://arxiv.org/pdf/2002.12177.pdf;
STiCA: https://arxiv.org/pdf/2103.10211.pdf;
AVMAE: https://arxiv.org/pdf/2212.05922.pdf;
XKD: https://arxiv.org/pdf/2211.13929.pdf;


**Limitations:**

In summary, this work lacks novelty, a comprehensive evaluation on different downstream benchmarks, and a thorough comparison with several prior works. These weaknesses outweigh the reasons to accept.

The proposed method has three objectives and all of them are heavily explored in the literature, and I do not see solid technical novelty. First, the masked reconstruction technique has already been implemented in several published works such as MAE, MAE_ST, VideoMAE, AudioMAE, CAV-MAE, and many more. Secondly, contrastive learning in multimodal/cross-modal setups has been widely studied. Lastly, the use of a frozen pretrained network to supervise an online encoder is not novel, whether in uni-modal or multi-modal setups.

---

> ### Author Rebuttal · Authors · 2023-08-10
>
> Dear Reviewer z1cD, thank you for reviewing our paper.
>
> Regarding the "limitation" about lacking comprehensive evaluation, please note that the focus of this work is on audio and audiovisual classification and we have reported extensive ablations and experiments in this field - and other reviewers agree with our assessment (e.g. dLj4 “Experiments and ablation studies are comprehensive”, kuAy - “very sufficient ablation study”). We are thankful for z1cD’s suggestion and included additional evaluations on EpicKitchen, UCF, HMDB, ... etc. While we do our best within the limited rebuttal period, we believe that the experiments reported already are extensive and more comprehensive than many published works.
>
> Regarding the “novelty” comment: We believe that MAViL's pre-training approach is novel and original (which is also recognized by RTTA and dLj4) and the resulting model is a technical contribution in of itself. For example, our multimodal student-teacher paradigm is entirely new; and we make one-step further to perform this in the aligned and contextualized space, leading to new SOTA results. We have tried to provide further ablations and hope these would convince you to re-evaluate MAViL's value for the audio and audiovisual community.
>
> We concur that MAVIL leverages contrastive learning and masked autoencoding, both of which are very broad concepts. If any work using these concepts is considered “lack of technical novelty”, then the works listed in the review would all categorize to lack novelty w.r.t. each other (MAE, VideoMAE, AudioMAE, SimCLR, Selavi, AVID, CVRL, BYOL, etc. are all forms of masked autoencoding as in MAE and contrastive learning as in MoCo).
>
> For your questions:
>
> **Q1: Why using (10s-A, 4s-V)? What’s the results of (4s-A, 4s-V)?**
> A1: Thank you for raising this point. The videos, along with their corresponding audio, have a duration of 10 seconds in AudioSet. As MBT, we default to using 8 frames in the visual branch. The choice ensures that the input token lengths remain comparable between the audio and video Transformers, with lengths of 512 and 768, respectively. In response to your valuable suggestion, we conducted experiments with various video spans (t_v) and audio spans (t_a) during the pre-training phase. The AS-20K fine-tuning results are:
>
> |(t_a, t_v)|(10,2)|(10,4)|(10,8)|(4,4)|(2,2)|
> | ----- | ------ | ------ | ----- | ------ | ------ |
> |Audio| 41.8 | 41.8 | 41.7| 40.9 |39.1|
> |Video| 24.9 | 24.8 | 24.5|24.7|24.8 |
>
> These findings indicate several key points: 1) It’s desirable to have a longer audio span (t_a=10). 2) A shorter video span (t_v=2) performs better, although the distinction is not substantial. 3) The overall performance remains considerably stable, whether utilizing asynchronous sampling or synchronous sampling (eg t_v=t_a=2).
>
> **Q2: The results w/o fusion encoder?**
>
> A2: Without the fusion encoder, the overall performance of MAViL will experience a notable decline (~1 mAP). We acknowledge your observation that the fusion encoder might, at first glance, appear to have a diminished role in Table 1a. However, it also serves an important role in MAViL stage-2 as it provides contextualized targets. Following your suggestion, we conducted an experiment that removes the encoder while keeping the other objectives constant. The “leave-fusion-out” results are:
>
> ||A|V|
> |---|---|---|
> |w/ fusion|41.8|24.8|
> |w/o fusion|40.7 (-1.1)|23.8 (-1.0)|
>
> The consistent drops validate the importance of fusion that exchanges complementary audio-video info for masked reconstruction.
>
> **Q3. Initializing w/ A-MAE and V-MAE for stage-2 training?**
>
> A3: Thank you for the suggestion and we request clarification. If you mean initializing the teacher model with A/V-MAE that generates prediction targets, then this ablation is already discussed in Table-4a column 2 (predicting A/V-MAE output), which significantly lags compared to using MAViL stage-1 as the teacher. This validates our intuition of using aligned and contextualized multimodal targets instead of heterogeneous uni-modal context (39.5->40.7(A),23.1->24.1(V)). If you mean student model initialization in stage-2, we experimented this and there is no change in the final mAP. Please let us know if this is consistent with your request.
>
> **Q4: Instead of 2-stage, what if using EMA? Why MSE and what if using cross-entropy or cosine distance?**
>
> A4: Thank you for your valuable suggestion. We did consider the utilization of EMA instead of stage-wise training during our initial investigation. However, unlike the uni-modal scenario (e.g, EMA in data2vec), we observed unstable training behavior when adopting EMA for student-teacher learning under the multimodal scenario.
>
> The instability is presumably a consequence of different convergence patterns in each modality.
> Consequently, this makes the design and tuning of mixture/update scheduler suboptimal (i.e., $\tau$ that controls the EMA update in  Data2vec Sec 3.3). Regrettably, this led to notable performance degradation:
>
> ||Stage-wise|EMA|
> |---|---|---|
> |A|41.8|40.3|
> |V|24.8|21.7|
>
> By employing a 2-stage training process, MAViL not only simplifies the methodology and achieves better performance, but it also mitigates the necessity of devising intricate update schedules, as was required in prior work.
>
> Additionally, the loss function comparison (MAViL-stage1) is as below:
>
> ||MSE|L1|BCE|cos-dist|
> |---|---|---|---|---|
> |A|39.0|38.5|38.3|36.9|
> |V|22.2|21.5|21.5|19.0|
>
> MSE is more robust. Please check our comment below for more discussion.
>
> **Q5. More information for Table 6?**
>
> Please check the pdf in the global response.
>
> **Q678. Linear Probing and additional fine-tuning experiments**
>
> Thank you for the suggestion. We conducted additional linear probing and fine-tuning experiments on EpicKitchen, UCF, HMDB, ESC, AS-20K. Tl;dr, MAViL achieved SOTA on audio and audiovisual classification tasks and comparable video performance with 8-frame inputs. Please check the pdf.

---

> > ### Author Response · Authors · 2023-08-11
> > **Additional discussion about choice of objective (Q4)**
> >
> > **Q4 Why MSE and what if using cross-entropy or cosine distance?**
> >
> > Regarding your question concerning the choice of loss function, we adhered to methodologies used in previous works such as MAE, VideoMAE, Aud-MAE to choose MSE as the reconstruction objective. MSE has demonstrated its efficacy in facilitating auto-encoding across continuous signals such as images, videos, and audio.
> >
> > With regards to your insightful suggestion of using other objectives such as binary-cross-entropy (BCE) and cosine distance, we carried out a comparison listed below (due to rebuttal time limit, we only compare MAViL-stage1 results):
> >
> > ||MSE|L1|BCE|cos-dist|
> > |---|---|---|---|---|
> > |A|39.0|38.5|38.3|36.9|
> > |V|22.2|21.5|21.5|19.0|
> >
> > As illustrated above, BCE also works but it is not as good as the widely used MSE in the context of masked auto-encoding. This disparity can be attributed to BCE's asymmetry, which introduces a bias towards a probability of 0.5, while MSE symmetrically regresses towards the exact input intensities, thereby enhancing performance.
> >
> > Furthermore, the attempt to minimize the cosine distance does not yield comparable outcomes in this context. This is likely due to the increased difficulty of reconstructing the precise input, which, in turn, compels the encoder to grasp intricate contextual elements like intensity, which is important for downstream classification tasks. Consequently, MSE results in a more robust self-supervisory process. In contrast, minimizing the cosine distance may inadvertently disregard intensity information. For instance, scenarios like a daylight scene and its corresponding darker or night-view might exhibit a small cosine distance between normalized patches. Failing to differentiate visual and spectrogram intensity may ultimately result in inferior SSL representations.

---

> > ### Comment · Reviewer_z1cD · 2023-08-21
> >
> > I thank the authors for attempting to address my comments, specifically for running the additional experiments. I believe this paper needs significant rework and I vote for rejection. Below are some additional comments in response to the authors' rebuttal.
> >
> > - The contribution of SimCLR/BYOL is significant in SSL and I **strongly disagree** that MAViL holds a similar level of contribution. In fact, I am surprised to find such comparisons.
> > - Apparently 10-sec audio and 2-sec video setup works best, then why did you use 4-sec video? using larger video is computationally expensive and not improving performance as well.
> > - Considering this is a multimodal framework, I would expect that it will work well on both modalities, if that's not the case there should be a specific reasoning. Without any solid reasoning, I am not convinced if "that the focus of this work is on audio and audiovisual " is justified.
> > - MAViL seems to struggle even outperforming the unimodal counterparts. First, it does not compare with VideoMAE. Second, even on audio downstream tasks (ESC50), it achieves just 0.3% better than AudioMAE, which is a fairly insignificant improvement considering the additional steps MAViL does compared to a simple AudioMAE.
> > - This paper lacks a thorough evaluation of video recognition (even after rebuttal) and does not make a fair comparison to the more recent works. In Table 4 (rebuttal pdf), the comparisons are incomplete, e.g., no comparison on VideoMAE, BEVT or other Transformer based arch., currently it only includes prior works based on R2plus1D (15M) vs their ViT-B (87M).
> > - No results on Kinetics400/600/700, SSv2; even the results on UCF101 and HMDB51 are fairly weak; E.g., VideoMAE reports 96.1 and 73.3 on UCF101 and HMDB51 pretrained on K400, while MAViL reports 90.5 and 60.7.

---

> ### Author Response · Authors · 2023-08-21
> **Thank you for your comment**
>
> We are grateful to receive your comment in the last 5 minutes before the discussion period ended. Due to time constraints, we would like to briefly respond:
>
> 1. We are not claiming a contribution such as SimCLR or BYOL. What we are stating is that with the reviewer’s lack of specific novelty claims, any work that does contrastive learning may be deemed having lack of novelty. MAViL has made its unique contributions with: 1) efficient contrastive learning with masking for both intra-modal and inter-modal context. 2) a novel self-training framework with multimodal MAE that predicts aligned and contextualized audio-video embeddings instead of raw signals in convention uni- and multi-modal MAEs. These to our best knowledge is novel and performing well.
>
> 2. The reviewer demands that our work outperforms all video classification methods, even though our focus is on audio-visual learning. This ask is similar to asking a work focusing on vision-language learning should outperform all image classification methods on ImageNet and similar datasets. Importantly, MAViL is trained on AudioSet under 8x224x224 for fair comparison to other baselines in Table 6. The requested video-recognition works are under a different visual footprint and on different pre-training datasets. We have tried our best to include the comparable (same footprint and same pre-training datasets) in the rebuttal pdf and show case MAViL outperforms them under the same setup.
> Further, the demand for being state-of-the-art on all types of tasks and benchmarks is an ask that clearly goes beyond what is required to publish in a top-tier conference.
>
> 3. Improvements on ESC: The improvement on top-1 acc is smaller in value compared to AudioSet since the baselines are already high (93+) in ESC. And MAViL outperforms AudMAE on AudioSet by a large margin.
>
> 4. No results on Kinetics400/600/700, SSv2? As mentioned, the focus on MAViL is on audio-visual classification instead of video-only task. Also the visual footprints and pre-training datasets are not comparable (see 2). We have included the comparable ones in the rebuttal pdf.
> In the experiments, MAViL has included sufficient comparison on 7 tasks, on the most common datasets in this field including AudioSet-20K, AudioSet-2M, VGGSound, VTT, YouCook, EpicKitchen, SPC, ESC … etc. And we also include additional experiments on UCF, HMDB per reviewer requested. With the limited amount of time in rebuttal, we are unable to include the experiments on all the datasets requested.

---

### Official Review · Reviewer_RTTA · 2023-07-02

**Soundness:** 4 excellent
**Presentation:** 4 excellent
**Contribution:** 3 good
**Rating:** 7
**Confidence:** 4

**Summary:**

The authors propose to learn audio-visual representations with two stage process with three self-supervision losses including 1. masked reconstruction 2. intra and inter-modal contrastive learning with masking 3. self-training via teacher-student setup to contrast on contextualized representations in 2nd stage. The authors show that proposed approach achieve state-of-the-art audio-visual classification performance on both AudioSet and VGGSound, audio-only tasks, and audio-to-video retrieval.

**Strengths:**

- The authors provide thorough ablation studies for each loss function introduced, masked ratio, choices of dataset size, backbone init, model size. This helps to understand the proposed system with a more complete view.
- The proposed approach to combine two popular loss function masked reconstruction and contrastive learning, and further introduce the self-training on contextualized representation is original.
- This paper is well written with clear narrative on the motivation and it provides sequences of experiments which help the reader to understand their thoughts step by step.

**Weaknesses:**

- This work lacks evaluation on the visual modality, especially video, as this work claims to be audio-video learner. Both AudioSet and VGGSound are mainly used in audio research, it would be beneficial to provide performance on some video benchmarks. For example, MBT also include EpicKitchens-100.
- Minor suggestion: consider add AS-20K in the caption of table 1, as it is less clear which dataset/task is used. Relevant to this, the use of AS-20K for ablation study is mentioned in the last paragraph of section 4.3, consider move it to the first paragraph of section 4.4 to better highlight this choice.

**Questions:**

- In section 4.3 it is mentioned that for downstream evaluation fine-tuning is applied. What is the performance on frozen encoders with linear probe if matching the output number of classes is necessary? Both fine-tuning and frozen can provide different perspectives of proposed system and can be used in different applications. It can provide a more complete view if provided.
- In table 4(b), what is the performance beyond three iterations, does it keep improving or the performance drops? It helps to include iteration 4 in either case.

**Limitations:**

- This work provides thorough study on proposed approach applied to audio related tasks including classification and crossmodal retrieval. It lacks some aspects on the video modality. It would be beneficial to include more evaluation with video downstream tasks.

---

> ### Author Rebuttal · Authors · 2023-08-09
>
> Dear reviewer RTTA,
>
> We appreciate your attention to recognizing the originality of MAViL's multimodal self-training approach, which leverages aligned and contextualized representations. Additionally, we thank you for recognizing our thorough model analysis and we will follow your editing suggestions. We address your comments and question as below:
>
> **W1: Video benchmark on Epic-Kitchen?**
>
> We thank you for this important suggestion. We experimented on Epic-Kitchen and summarize the top-1 acc for verbs, nouns and actions in below:
>
> ||Type|Noun|Verb|Action|
> |---|---|---|---|---|
> |A-SlowFast|A|22.8|46.5|15.4|
> |MBT|A|22.4|44.3|13.0|
> |MAViL|A|26.8|53.1|20.2|
> |SlowFast|V|50.0|65.6|38.5|
> |MTV|V|60.5|67.8|46.7|
> |MBT|V|56.4|62.0|40.7|
> |MAViL|V|54.8|69.4|45.1|
> |MBT|A+V|58.0|64.8|43.4|
> |MAViL|A+V|56.6|71.8|46.0|
>
> MAViL (A,A+V), outperforms MBT(A,A+V) on Noun and Action in Epic-Kitchen. Note that both MTV and MBT exploited ImageNet-21K (14M) supervision while MAViL is fully self-supervised on AS-2M. And MAViL uses 8-frame video inputs compared to 32-frame in MTV. MAViL encodes only 8 frames to maintain a comparable sequence lengths between the audio (512) and video (768) branches. Additional UCF and HMDB fine-tuning results can be found in the pdf attached in the general response.
>
> **Q1: Linear probing results and more downstream tasks?**
>
> A1: Following the suggestion from you and other reviewers, we conducted additional audio probing on ESC and AS-20K and visual probing on HMDB51, UCF101:
>
> ||PT-Data|ESC-FT|ESC-Linear|AS-20K-FT|AS-20K-Linear|
> |---|---|---|---|---|---|
> |XDC|A+V,AS|-|84.8|-|-|
> |AVID|A+V,AS|-|89.1|-|-|
> |BraVe|A+V,AS|-|90.4|-|-|
> |CrissCross|A+V,AS|-|90.5|-|-|
> |SS-AST|A,AS|88.8|85.6|31.0|-|
> |A-MAE|A,AS|94.1|89.5|37.0|-|
> |MAViL|A+V,AS|94.4|90.8|41.8|30.0|
>
> ||PT-Data|#frms|U101-Linear|U101-FT|H51-Linear|H51-FT|
> |---|---|---|---|---|---|---|
> |XDC|A+V,IG65M|8|-|84.9|-|48.8|
> |AVID|A+V,AS|8|-|88.6|-|57.6|
> |CrissCross|A+V,AS|8|87.7|89.4|56.2|58.3|
> |MAViL|A+V,AS|8|89.1|90.5|58.3|60.7|
> |BraVe*|A+V,AS|32*|93.0|95.6|69.4|76.5|
>
> Note that the BraVe is with high temporal resolution (multiple-views of 128/32 frames, while 8 frames for MAViL). MAViL does not claim SOTA on action recognition tasks because of the visual footprints we chose (8 frame under 2 fps, 4 second video) and simplified visual augmentation.
>
> MAViL focused more on the audio and audio-video classification performance and targeted on the hardest task on AudioSet to compared with other audio and audio-video baselines in Table 6. With the page limit constraint, we were unable to include dedicated visual results and analysis. We will include video-related results and analysis in the additional page granted in the final version.
>
>
> **Q2: Performance improvement in multimodal self-training beyond 3rd iteration?**
>
> A2: Thank you for bringing this up. The table below extends Table 4b and summarizes the full AS-20K results of iteration 0-5 in multimodal self-training.
>
> |Iter.|None|1|2|3|4|5|
> |---|---|---|---|---|---|---|
> |A|39.0|40.7|41.5|41.8|41.9|41.9|41.8|
> |V|22.2|24.1|24.6|24.8|25.0|24.8|24.9|
>
> Self-training for more iteration provides marginal gain after the 3rd/4th iteration. We choose to train 3 iterations for simplicity. We will adjust the presentation in Table 4(b) to include these results.

---

> > ### Comment · Reviewer_RTTA · 2023-08-20
> >
> > Thank the authors for your responses and extra experiments. I would suggest integrating the results of linear probe and iteration beyond 3rd into the final paper if possible and other video related results into additional pages. I think this is a good work and I am maintaining my rating.

---

> ### Author Response · Authors · 2023-08-21
> **Thank you for the comment**
>
> We sincerely appreciate the time you dedicated to reviewing our paper. We are delighted that your inquiries have found satisfactory answers, and we are committed to integrating your valuable suggestions into the paper's final version with the additional page granted. If any specifics are unclear, we would be more than willing to address them. Thank you.

---

### Official Review · Reviewer_kuAy · 2023-07-09

**Soundness:** 4 excellent
**Presentation:** 4 excellent
**Contribution:** 4 excellent
**Rating:** 8
**Confidence:** 4

**Summary:**

This paper presents a new self-supervised learning framework for audio-visual modalities. It combines three types of SSL techniques, namely MAE, contrastive learning, and student-teacher distillation. The proposed method is a two-stage process by first training a teacher network to reconstruct the raw signal and then using this network to guide a student network to learn in the second stage. Authors show that this method improves the SOTA on AudioSet, VGG-Sounds as well as several other classification and retrieval benchmarks.

**Strengths:**

Although the three SSL losses in this paper are not new, this paper did a good job of putting them together and making them work. The performance improvement over the SOTA method is also pretty decent, validating the effectiveness of the approach.

The authors conducted a very sufficient ablation study of the model architecture, showing that multi-modal fusion, contrastive learning, reconstruction, and distillation all contribute to the model's performance.

The writing is very clear-written and easy to follow.

**Weaknesses:**

In the implementation details, authors mentioned many data augmentations are used in training. How much do they matter? I do not see an ablation on this component.

The limitation of this work is not discussed in this paper.

**Questions:**

I'd appreciate the authors' response on how much data augmentation matters and the potential limitation of this work.


====post rebuttal===
The authors' response addressed my concerns and I am keeping the original score.

**Limitations:**

Limitations and societal impact are not discussed in this work.

---

> ### Author Rebuttal · Authors · 2023-08-09
>
> We thank you for your review of our manuscript and for acknowledging MAViL’s SOTA performance accompanied by a comprehensive analysis. With regard to your comments and questions please see below:
>
> **Q1: Impact of data augmentation?**
>
> A1: Thank you for the great question! Comprehensive details of the augmentation techniques and configurations can be referenced in Table 1 in Supplementary. In the pretraining phase, MAViL exclusively employs audio/video cyclic rolling. In the fine-tuning phase, standard data augmentation methods, including SpecAug, Mixup, magnitude augmentations, and others that have been utilized in prior studies (such as AST, CAV-MAE, Aud-MAE in Table 6), are incorporated.
>
> For your question, we conducted an experiment to evaluate the impact of data augmentation by removing all audio/video augmentation. The results are outlined below:
>
> || AS-20K (V) | AS-20K (A)| AS-20K (A+V) |
> | -----  | ------ | ------ | ------ |
> |MAViL w/ aug | 41.8 | 24.8 | 44.9 |
> |MAViL w/o PT aug | 41.6 | 24.7 | 44.7 |
> |MAViL w/o PT and FT aug | 39.1 | 23.3 | 42.5 |
>
> The results show that data augmentation plays a less significant role in pre-training, and is relatively important for fine-tuning. Please note that previous SOTA models (eg. MBT, CAV-MAE, AudMAE) are all using augmentations.
>
> **Q2: Limitation of this work?**
>
> A2: With the page limits, we have addressed MAViL’s limitations in the Supplementary materials. The main constraint of MAViL lies in its utilization of AudioSet (2M audio-video recordings) for audio-visual pre-training. This choice raises several concerns:
> 1. Smaller dataset size: MAViL required audio-video data for pre-training and AudioSet is currently the best dataset available. Compared to datasets like ImgNet-21K (with 14M images) used in MBT, AudioSet's size is notably smaller.
> 2. Short recordings: Each recording in AudioSet is around 10 seconds. This poses challenges when applying MAViL to tasks such as automatic speech recognition (ASR), where extended audio sequences are common.
> 3. Weak Audio-Video correspondence: The in-the-wild recordings present in AudioSet exhibit limited audio-video correspondence, posing difficulties in achieving robust audio-visual learning.
>
> We anticipate that MAViL's performance will see further enhancements with the availability of larger-scale audio-video datasets.

---

### Author Rebuttal · Authors · 2023-08-10

Dear Reviewers and Area Chairs,

We thank you for the time you invested in reviewing our paper and providing invaluable suggestions that have significantly improved the quality of MAViL. In particular, we wish to express our heartfelt thanks to RTTA and dLj4 for acknowledging the originality and substantial contribution of MAViL's multimodal self-training approach that reconstructs aligned and contextualized representations without reliance on external models or data sources. Furthermore, we extend our thanks to kuAy, RTTA, and dLj4 for acknowledging our thorough model analysis, comprehensive study, and state-of-the-art performance in audio and audiovisual classification and retrieval tasks across seven datasets. Finally, we are grateful for the valuable comments and questions raised by z1cD, kuAy, RTTA and dLj4. Please find the individual rebuttal for our detailed response.

Enclosed within the attached PDF file, we present supplementary fine-tuning and linear probing experiments conducted on EpicKitchen, UCF101, HMDB51, ESC, and AudioSet-20K, in response to reviewer’s requests of additional experiments.

We thank you for your review. Please let us know if you have any remaining concerns and we are happy to address them.

---

### Decision · Program_Chairs · 2023-09-21

**Decision:**

Accept (poster)

**Comment:**

This is a borderline submission. Two reviewers argued that the strong empirical results are sufficient for acceptance. The other two reviewers found that the empirical evaluation is not sufficient for such claims, and expected to see improvements to the unimodal baselines and better positioning with respect to substantial amount of related work. In the rebuttal the authors provided additional empirical evidence which suggests which strengthened the main claims and clarifies the impact of individual loss components. I will recommend acceptance of this submission given strong empirical performance.